# The Expanding Genetic Architecture of Arteriopathies: From Canonical TAAD Genes to Emerging Connective Tissue and Signaling Pathways

**DOI:** 10.3390/medsci13030155

**Published:** 2025-08-25

**Authors:** Luke Dreher, Hussein Abdul Nabi, Hunter Vandolah, Stephen Brennan, George Bcharah, Hend Bcharah, Mayowa A. Osundiji, Linnea M. Baudhuin, Fadi E. Shamoun

**Affiliations:** 1Department of Cardiovascular Medicine, Mayo Clinic Arizona, 5777 East Mayo Blvd, Phoenix, AZ 85054, USA; dreher.luke@mayo.edu (L.D.); abdulnabi.hussein@mayo.edu (H.A.N.); vandolah.hunter@mayo.edu (H.V.); bcharah.george@mayo.edu (G.B.); hend.bcharah@mayo.edu (H.B.); 2Department of Cardiology, University of Galway, H91 TK33 Galway, Ireland; stephen.o.brennan@universityofgalway.ie; 3Department of Clinical Genomics, Mayo Clinic Arizona, Phoenix, AZ 85054, USA; osundiji.mayowa@mayo.edu; 4Department of Laboratory Medicine and Pathology, Mayo Clinic, Rochester, MN 55905, USA; baudhuin.linnea@mayo.edu; 5Department of Clinical Genomics, Mayo Clinic, Rochester, MN 55905, USA

**Keywords:** thoracic aortic aneurysm, thoracic aortic dissection, heritable aortapathy, TGF-β signaling, extracellular matrix remodeling

## Abstract

Thoracic aortic aneurysm and dissection (TAAD) encompasses a clinically heterogeneous group of diseases characterized by high morbidity and mortality. Genetic studies over the past two decades have significantly expanded our understanding of the molecular landscape underlying heritable TAAD, revealing contributions from multiple interconnected biological pathways. This review systematically summarizes more than 75 genes implicated in TAAD pathogenesis, categorizing them according to major mechanistic roles including TGF-β signaling, extracellular matrix remodeling, smooth muscle cell contractility and cytoskeletal regulation, cell–matrix and cell–cell adhesion, metabolic processes, ion transport, and transcriptional regulation. Special emphasis is placed on emerging genes with variable or overlapping clinical phenotypes, dual-mechanism candidates, and their implications for personalized clinical management, including surveillance and surgical intervention thresholds. The integration of molecular insights into clinical practice, along with cautious consideration of genes of uncertain significance, promises to enhance diagnostic precision and risk stratification in individuals and families affected by heritable aortic disease.

## 1. Introduction

Thoracic aortic aneurysm and dissection (TAAD) is a clinically silent, yet potentially catastrophic condition characterized by progressive weakening of the aortic wall [1]. Often asymptomatic until acute dissection or rupture, TAAD carries a high burden of morbidity and mortality, particularly when diagnosis is delayed or familial risk is unrecognized. Historically, the understanding of heritable TAAD was anchored in syndromic disorders such as Marfan syndrome (caused by pathogenic *FBN1* variants) and LDS (linked to *TGFBR1*, *TGFBR2*, *SMAD3*, *SMAD2*, *TGFB2*, and *TGFB3*), conditions defined by clear Mendelian inheritance patterns and systemic features [2]. Genetic causes for nonsyndromic TAAD have been identified more recently and include genes such as *ACTA2, MYLK*, and *MYH11*.

Recent advances have broadened the genetic landscape of TAAD beyond classical syndromic and nonsyndromic conditions. Research now implicates diverse biological pathways, including transforming growth factor-beta (TGF-β) signaling, vascular smooth muscle cell (VSMC) contractility, extracellular matrix (ECM) structure and remodeling, proteoglycan biosynthesis, and transcriptional regulation. Emerging mechanisms such as somatic mosaicism, polygenic inheritance, and epigenetic modulation are redefining how genetic risk is assessed in patients lacking overt syndromic features. (An overview of how these pathways intersect to influence TAAD pathogenesis is presented in Figure 1, providing a framework for the detailed discussions in subsequent sections).

Large-scale genomic studies, including the Million Veteran Program genome-wide association study [4], have highlighted the role of common variant burden and novel risk loci in nonsyndromic aortopathy. Similarly, analysis of mosaicism in nonsyndromic patients has revealed that low-level somatic variants in canonical TAAD genes (*FBN1* and *NOTCH3*) account for a subset of early-onset aortopathy [5]. Sex-specific differences in dissection risk have also been identified, with a large female-focused cohort revealing unique genetic architecture and chromosomal associations [6]. Epigenetic studies have further expanded this landscape of disease modifiers, with differential methylation, histone signatures, and noncoding RNAs increasingly recognized as contributors to aortic wall vulnerability [7].

Despite these advances, no unified framework exists to organize the over 75 genes implicated in TAAD and related arteriopathies. This review aims to address this gap by synthesizing and expanding the genetic architecture of TAAD, ranging from canonical syndromic genes to underrecognized contributors and molecular modifiers. By integrating classical Mendelian syndromes with insights from next-generation sequencing, variant curation frameworks, and cellular signaling networks, we propose a pathway-based model to enhance diagnostic precision, risk stratification, and tailored surveillance in clinical care.

## 2. Classical Monogenic Syndromes and TAAD

A foundational understanding of heritable thoracic aortic disease (HTAD) stems from classical Mendelian syndromes [8,9,10]. These include Marfan syndrome, LDS, and vascular Ehlers–Danlos syndrome (vEDS), which represent the earliest recognized genetic contributors to TAAD and remain central to clinical genetic testing panels.

Marfan syndrome, the prototypical heritable aortopathy, results from pathogenic variants in *FBN1*, which encodes fibrillin-1, a critical ECM glycoprotein involved in microfibril assembly and sequestration of latent TGF-β complexes. Loss-of-function or dominant-negative *FBN1* variants lead to aortic root aneurysms, skeletal overgrowth, lens dislocation, and dural ectasia [2]. Inheritance is autosomal dominant, with variable expressivity. Aortic dissection in Marfan syndrome often occurs independently of hypertension, underscoring the underlying structural vulnerability.

LDS encompasses a group of disorders caused by variants in genes encoding components of the TGF-β signaling cascade, including *TGFBR1*, *TGFBR2*, *SMAD3*, *TGFB2*, *TGFB3*, and *SMAD2* [11,12,13]. LDS is characterized by aggressive arterial disease with aneurysms and dissections occurring throughout the arterial tree, often at smaller diameters and younger ages than Marfan syndrome. Additional features may include bifid uvula, hypertelorism, craniosynostosis, and skeletal deformities [2]. Most cases are autosomal dominant and frequently arise de novo. While aortic root dilation is common, LDS is notable for widespread aneurysms and dissections affecting the cervical, visceral, and iliac arteries. LDS subtypes (I–V) reflect the specific gene affected but share core pathophysiologic mechanisms involving dysregulated TGF-β signaling and paradoxical pathway activation.

vEDS is caused by pathogenic variants in *COL3A1*, which encodes type III procollagen. This autosomal dominant disorder predominantly affects medium-sized arteries and visceral vessels, resulting in spontaneous rupture of arteries, hollow organs, or the uterus. Aortic involvement may occur but is less predictable. Hallmark findings include skin translucency, easy bruising, and characteristic facial features [2,14]. Although vEDS is the main type of EDS with vascular fragility, it is becoming increasingly recognized that other forms of EDS may also be predisposed to aneurysm and rupture [15].

Other genetic causes of syndromic TAAD include Shprintzen–Goldberg syndrome (SGS) (*SKI*), congenital contractural arachnodactyly (*FBN2*), and autosomal recessive cutis laxa (*EFEMP2*, *ELN*) [16,17,18]. These conditions demonstrate overlapping features with Marfan or LDS but are distinguished by gene-specific developmental anomalies and systemic findings. Arterial tortuosity syndrome, caused by recessive variants in *SLC2A10*, results in widespread tortuosity, aneurysms, and craniofacial dysmorphisms.

The common thread across these syndromes is a profound disruption in connective tissue structure, ECM signaling, or vascular smooth muscle integrity. These early models have shaped both the clinical suspicion for genetic testing and the development of management guidelines [19], including reduced surgical thresholds for aortic repair and lifelong vascular surveillance. For instance, repair is often recommended at diameters as low as 4.0–4.5 cm in Loeys–Dietz syndrome and some *COL3A1* variants compared to 5.0–5.5 cm in nonsyndromic cases [20].

Importantly, not all patients with these syndromes meet strict clinical criteria or display overt extra-aortic features [2]. Genotype-first approaches have revealed a spectrum of phenotypes even within single-gene disorders. For example, some *FBN1* variants cause isolated TAAD without skeletal involvement, while *SMAD3* variants can present with thoracic aneurysm alone or in the context of Aneurysms–Osteoarthritis syndrome (AOS). This phenotypic variability underscores the need to move beyond eponymous syndromes toward a mechanistic and pathway-level classification of heritable arteriopathies.

## 3. Functional Framework for TAAD Genes

The molecular pathways underlying heritable TAAD are not discrete but form an interconnected network in which dysregulation in one domain can propagate pathologic changes in others. For example, loss-of-function variants in TGF-β receptors reduce canonical signaling fidelity yet paradoxically elevate downstream pSMAD2 activity, driving excessive extracellular matrix turnover. ECM fragmentation, in turn, alters mechanotransduction, disrupting focal adhesion complexes and cytoskeletal stability. These structural changes impair vascular smooth muscle cell contractility, destabilize ion homeostasis, and promote a synthetic phenotype characterized by increased matrix-degrading enzyme activity. Metabolic and redox regulators further modulate these cascades by influencing methylation patterns, protein folding, and oxidative stress responses. Together, these processes establish a self-reinforcing cycle in which signaling imbalance, matrix disorganization, mechanical weakness, and altered cellular physiology collectively accelerate aneurysm progression and dissection risk (refer to Figure 1).

### 3.1. TGF-β and SMAD Signaling

#### 3.1.1. Core Pathway Overview

Dysregulation of the TGF-β signaling pathway is among the most well-established mechanisms underlying TAAD. This pathway controls vascular homeostasis by regulating ECM composition, SMC phenotype, proliferation, and inflammatory tone. Central to this signaling cascade is a tightly regulated axis involving ligands, receptors, intracellular mediators, and feedback inhibitors that function in coordination to preserve aortic wall structure and adaptability (refer to Figure 2, Table 1).

**Table 1 medsci-13-00155-t001:** Comprehensive functional and clinical classification of genes implicated in heritable TAAD.

Gene	Protein	Function	Location	Pathway Association	sTAAD	nsFTAAD	Phenotype	AA Size (cm) for SI	MoA	SoA	Category (A–D)	OMIM	Sources
*ABCC6*	ATP-binding cassette, subfamily C, member 6	ATP-binding cassette, serves as an export pump of unknown molecules that inhibit pathological calcification	16p13.11	Metabolic and Ion Transport	No	No	Pseudoxanthoma elasticum (PXE); arterial calcification, ECM fragmentation, increased stiffness	N/A	AD and AR	N/A	N/A	603234	[20,21,22,23,24]
*ACTA2*	Smooth muscle alpha-actin	Contractile protein in vascular smooth muscle cells	10q23.31	Smooth Muscle Contraction/Mechanotransduction	Yes	Yes, most common cause	Aortic aneurysm and dissection (ascending), premature CAD, BAV, stroke	4.5–5.0 cm	AD	Definitive	A2	611788 613834 614042	[20,21,23,24,25]
*ACTN2*	Alpha-actinin-2	Crosslinks actin filaments at the Z-disc, modulates myofibrillar tension	1q43	Smooth Muscle Contraction/Mechanotransduction	No	Possibly	Left-dominant arrhythmogenic cardiomyopathy; suggested vascular overlap	N/A	AD	N/A	N/A	102573	[20,21,23,24,26]
*ADAMTS2*	Procollagen 1 N-proteinase	Removes propeptides to allow for fibril self-assembly	5q55.3	Extracellular and Structural Matrix	Yes—EDS dermatosparaxis type (EDS VIIC)	No	Severe skin fragility, joint hypermobility, connective tissue disorder	N/A	AR	N/A	N/A	604539	[20,21,23,24,27]
*ATP6V0A2*	ATPase, H+ transporting, lysosomal, V0 subunit A2	Subunit of a vacuolar-type proton pump that acidifies endosomes, lysosomes, Golgi apparatus	12q24.31	Extracellular and Structural Matrix	Yes—cutis laxa and wrinkly skin syndrome (hypothesized to affect vasculature)	No	Connective tissue laxity, skin wrinkling, theoretical vascular remodeling impairment	N/A	AR	N/A	N/A	611716	[20,21,23,24,28]
*ATP7A*	ATPase, Cu2+-transporting alpha polypeptide	Transmembrane copper-transporting P-type ATPase supports lysyl oxidase function to crosslink collagen/elastin	Xq21.1	Metabolic and Ion Transport	Yes—Menkes disease, occipital horn syndrome	No	Arterial tortuosity, dissection-prone vasculature, connective tissue fragility	N/A	XLR	N/A	N/A	300011	[20,21,23,24,29]
*B3GALT6*	Beta-1,3-galactosyltransferase 6	Facilitates synthesis of heparan sulfate and chondroitin sulfate (glycosaminoglycan matrix proteins)	1p36.33	Extracellular and Structural Matrix	Yes—spondylodysplastic EDS (spEDS)	No	Short stature, joint hypermobility, skin fragility, occasional vascular involvement	N/A	AR	N/A	N/A	615291	[20,21,23,24,30]
*B4GALT7*	Beta-1,4-galactosyltransferase 7	Facilitates synthesis of heparan sulfate and chondroitin sulfate (glycosaminoglycan matrix proteins)	5q35.3	Extracellular and Structural Matrix	Yes—spondylodysplastic EDS type 1	No	Joint dislocations, short stature, limb anomalies, possible vascular fragility	N/A	AR	N/A	N/A	604327	[20,21,23,24,31]
*BGN*	Biglycan	Small leucine-rich proteoglycan, which binds to collagen fibrils and modulates ECM stiffness and profibrotic signaling	Xq28	Overlap/Dual-Phenotype/Emerging Candidates	Yes, Meester–Loeys syndrome (X-linked, with vascular features)	No	Aortic dilation, reduced collagen content, increased pSMAD2, occasional aortic rupture	N/A	XL	N/A	N/A	301870	[20,21,23,24,32]
*CBS*	Cystathionine beta-synthase	Catalyzes the first irreversible step of transsulfuration, leading to the production of homocysteine	21q22.3	Metabolic and Ion Transport	Yes—homocystinuria	No	Endothelial dysfunction, thrombosis, ECM degradation due to hyperhomocysteinemia	N/A	AR	Limited	C	613381	[20,21,23,24,33]
*CDH2*	Cadherin 2	Neuronal-cadherin is also involved in cell–cell adhesion between smooth muscle cells	18q12.1	Cell–Cell Junction/Adhesion	No	Possibly	arrhythmogenic right ventricular cardiomyopathy (ARVC); ocular and genital defects; cardiac anomalies	N/A	AD	N/A	N/A	114020	[20,21,23,24,34]
*CHST14*	Carbohydrate sulfotransferase 14	Catalyzes 4-O-sulfation of dermatan sulfate (stereoisomer of chondroitin sulfate)	15q15.1	Extracellular and Structural Matrix	Yes—musculocontractural EDS (EDSMC1)	No	Arterial dilation, aneurysm, or dissection reported in EDSMC1	N/A	AR	N/A	N/A	608429	[20,21,23,24,35]
*COL12A1*	Type XII collagen	Fibril-associated collagen with interrupted triple helices (FACIT) tethers fibrils to surrounding structures	6q13-q14.1	Extracellular and Structural Matrix	No	No	myopathic EDS, skeletal/connective abnormalities; no aortic findings	N/A	AD	N/A	N/A	120320	[20,21,23,24,36]
*COL1A1*	Type I collagen, alpha-1 chain	Alpha-1 subunit of type 1 collagen	17q21.33	Extracellular and Structural Matrix	Yes—arthrochalasia-type EDS and classic EDS (rare vascular involvement)	No	Hyperextensible skin, joint laxity, early aortic valve disease, possible borderline aortic dilation	N/A	AD	No Evidence	D	120150	[14,20,21,23,24,37]
*COL1A2*	Type I collagen, alpha-2 chain	Alpha-2 subunit of type 1 collagen	7q21.3	Extracellular and Structural Matrix	Yes—cardiac valvular EDS	No	Mitral/aortic valve disease, borderline root dilation, joint hypermobility	5.0 cm	AD and AR	No Evidence	D	120160	[14,20,21,23,24,37]
*COL3A1*	Type III collagen, alpha-1 chain	Dominant type 3 fibrillar collagen	2q32.2	Extracellular and Structural Matrix	Yes—vascular Ehlers–Danlos syndrome (vEDS)	No	Arterial rupture/dissection, thin translucent skin, easy bruising, uterine/intestinal rupture	5.0 cm	AD	Definitive	A1	120180	[14,20,21,23,24]
*COL5A1*	Type V collagen, alpha-1 chain	Regulates collagen fibril assembly and ECM integrity; modulates the diameter of collagen I fibrils	9q34.3	Extracellular and Structural Matrix	Yes—classic EDS	No	Joint hypermobility, atrophic scarring, fragile skin; potential mild aortic dilation	Standard	AD	No Evidence	D	120215	[14,20,21,23,24,37]
*COL5A2*	Type V collagen, alpha-2 chain	Alpha-2 subunit of type 5 collagen	2q32.2	Extracellular and Structural Matrix	Yes—Ehlers–Danlos syndrome, classic type 2	No	Classic EDS (joint hypermobility, skin hyperextensibility), theoretical aortic impact	Standard	AD	No Evidence	D	120190	[14,20,21,23,24,37]
*COLGALT1*	Collagen beta (1-O) galactosyltransferase 1	Glycoyslates hydroxylysine residues, stabilizing triple-helix formation	19p13.11	Extracellular and Structural Matrix	No	No direct evidence; potential role in ECM disorder	Defects in collagen IV trafficking; no confirmed aortic phenotype	N/A	AR	N/A	N/A	617531	[20,21,23,24,38]
*CTNNA1*	Catenin alpha 1	Mediates adherens junction linkage to the actin cytoskeleton	5q31.2	Overlap/Dual-Phenotype/Emerging Candidates	No	Possibly	Germline inactivating variants predispose to early-onset diffuse gastric cancer, with some carriers also developing invasive lobular breast cancer; loss impairs cell–cell adhesion, alters focal adhesion dynamics, increases cell migration, and affects epithelial tissue integrity	N/A	AD	N/A	N/A	116805	[20,21,23,24,39,40]
*CTNNB1*	Catenin beta 1	Mediates adherens junction linkage to the actin cytoskeleton	3p22.1	Overlap/Dual-Phenotype/Emerging Candidates	No	Possibly	Global developmental delay, intellectual disability (often moderate–severe), speech impairment, microcephaly, hypotonia, spastic diplegia; implicated in heart disease (e.g., dilated cardiomyopathy) and cardiac development	N/A	AD	N/A	N/A	116806	[20,21,23,24,39,40]
*DCN*	Decorin	Small leucine-rich proteoglycan that regulates collagen fibrillogenesis by controlling fibril diameter and spacing	12q21.33	Overlap/Dual-Phenotype/Emerging Candidates	No	Possibly	Aortic aneurysm due to extracellular matrix disruption (based on role in proteoglycan regulation)	N/A	AD	N/A	N/A	125255	[20,21,23,24,41]
*DSE*	Dermatan sulfate epimerase	Converts D to L-glucuronic acid during dermatan sulfate synthesis	6q22.1	Extracellular and Structural Matrix	Yes—musculocontractural Ehlers–Danlos syndrome (mcEDS) with potential vascular features in severe cases	No	Ehlers–Danlos syndrome, musculocontractural type 2	N/A	AR	N/A	N/A	605942	[20,21,23,24,42]
*DSP*	Desmoplakin	Core desmosome component, tethering intermediate filaments between adjacent cells to maintain cohesion during cyclic motion	6p24.3	Cell–Cell Junction/Adhesion	Yes, carvajal/naxos-like syndromes	No	Dilated cardiomyopathy, left ventricular noncompaction, arrhythmogenic cardiomyopathy with variable vascular involvement	N/A	AD and AR	N/A	N/A	125647	[20,21,23,24,43]
*ELN*	Elastin	Core component of the elastic fiber	7q11.23	Extracellular and Structural Matrix	Yes—Williams–Beuren syndrome and cutis laxa	Yes—reported in isolated SVAS and aortic aneurysms	Supravalvular aortic stenosis (SVAS), cutis laxa, arterial tortuosity, aneurysms, elastic fiber fragmentation	Standard	AD	Limited	B2	130160	[20,21,23,24,44]
*EMILIN1*	Emilin 1	Elastic fiber-associated glycoprotein regulating TGF-β precursor maturation	2p23.3	TGF-β/SMAD Signaling	No	Proposed only	In mouse models, EMILIN1 deficiency results in aortic root aneurysms, likely driven by dysregulated TGF-β signaling	N/A	AR	N/A	N/A	130660	[20,21,23,24,45]
*ENPP1*	Ectonucleotide pyrophosphatase/phosphodiesterase 1	Generate inorganic pyrophosphate, which inhibits vascular calcification	6q23.2	Metabolic and Ion Transport	Yes—generalized arterial calcification of infancy (GACI1)	No	Arterial calcification, periarticular and aortic mineralization, PXE-like features	N/A	AR	N/A	N/A	173335	[20,21,23,24,46]
*FBLN4 (EFEMP2)*	EGF-containing fibulin-like extracellular matrix protein 2	Promote elastic fiber crosslinking and stability between fibrillin, elastin, and lysyl oxidases	11q13.1	Extracellular and Structural Matrix	Yes—AR cutis laxa type IB	No	Severe cutis laxa, ascending aortic aneurysm, arterial tortuosity, pulmonary artery hypoplasia	Standard	AR	Moderate	B1	604633	[20,21,23,24,47]
*FBLN5*	Fibulin5	Promote elastic fiber crosslinking and stability between fibrillin, elastin, and lysyl oxidases	14q32.12	Extracellular and Structural Matrix	Proposed—cutis laxa (AR or AD forms), but no confirmed aortic aneurysms in human cases	No	Elastic fiber defects, tortuosity, emphysema, pelvic organ prolapse (in mice); no clear human aortic aneurysm phenotype	N/A	AD and AR	N/A	N/A	604580	[20,21,23,24,47]
*FBN1*	Fibrillin 1	Central microfibril component, scaffolds elastic fibers, and sequesters latent TGF-β complexes	15q21.1	Extracellular and Structural Matrix	Yes—Marfan syndrome	Yes	Aortic root aneurysm, ectopia lentis, long limbs, scoliosis, pectus deformity, joint laxity, skin striae	5.0 cm	AD	Definitive	A1	134797	[20,21,23,24,48]
*FBN2*	Fibrillin 2	Fetally expressed microfibril scaffold for elastin	5q23.3	Extracellular and Structural Matrix	Yes—congenital contractural arachnodactyly (CCA)	No	Joint contractures, arachnodactyly, scoliosis, “crumpled” ears; mild aortic dilation occasionally noted	Standard	AD	N/A	N/A	612570	[16,20,21,23,24]
*FLCN*	Folliculin	Tumor suppressor, influencing mTOR signaling and SMC turnover behavior	17p11.2	Transcriptional/Nuclear Envelope Regulation	No	No	Hypothesized role in ECM remodeling and SMC behavior	N/A	AD	N/A	N/A	607273	[20,21,23,24,49]
*FLNA*	Filamin A	Actin-binding protein critical for cytoskeleton structure and signaling	Xq28	Smooth Muscle Contraction/Mechanotransduction	Yes, periventricular nodular heterotopia syndrome with aortic dilation	No	Aortic aneurysm with connective tissue and neurologic features (PNH)	Standard	X-linked Dominant	Limited	B1	300017	[20,21,23,24,50]
*FLNC*	Filamin C	Actin binding and crosslinking in muscle cells, cytoskeletal structure reshaping	7q32.1	Smooth Muscle Contraction/Mechanotransduction	No	Possible, based on altered expression	Skeletal and cardiac myopathies; possible VSMC cytoskeletal effects	N/A	AD	N/A	N/A	102565	[20,21,23,24,51]
*GJA4*	Gap junction protein, alpha-4 (connexin 37)	Facilitate ionic and metabolic exchange between endothelial cells and endothelium/SMCs	1p34.3	Overlap/Dual-Phenotype/Emerging Candidates	No	Possibly	Disruption in GJA5 may impair vascular tone regulation and promote ECM remodeling. It has also been associated with essential hypertension and vessel wall dysfunction, features that could plausibly contribute to aneurysmal susceptibility, though not directly proven	N/A	N/A	N/A	N/A	121012	[20,21,23,24,40,52]
*GJA5*	Gap junction protein, alpha-5 (connexin 40)	Facilitate ionic and metabolic exchange between endothelial cells and endothelium/SMCs	1q21.2	Overlap/Dual-Phenotype/Emerging Candidates	No	Possibly	Connexin-40 dysregulation; may influence vascular tone, but no direct evidence of TAAD	N/A	AD	N/A	N/A	121013	[20,21,23,24,40,52]
*HEY2*	HES-related family bHLH transcription factor with YRPW Motif 2	Helix–loop–helix transcription factor involved in early embryonic vessel formation and downstream Notch signaling	6q22.31	Transcriptional/Nuclear Envelope Regulation	No	Yes	Associated with congenital heart disease and thoracic aneurysms	N/A	Mixed	N/A	N/A	604674	[20,21,23,24,53]
*HNRNPK*	Heterogeneous nuclear ribonucleoprotein K	DNA- and RNA-binding protein involved in chromatin remodeling, transcription, mRNA splicing and stability, translation	9q21.32	Transcriptional/Nuclear Envelope Regulation	No	No	possibly influences SMC contractility and response to strain	N/A	AD	N/A	N/A	600712	[20,21,23,24,54,55]
*IPO8*	Importin-8	Nuclear transport protein involved in TGF-β signaling regulation	12p11.21	TGF-β/SMAD Signaling	Yes (associated with syndromic features including arterial tortuosity and aneurysm)	No	Aneurysms, arterial tortuosity, possibly other syndromic traits	N/A	AR	N/A	N/A	605600	[20,21,23,24,56,57]
*JAG1*	Jagged1	Notch receptor ligand, facilitating signaling leading to embryonic vascular development and vascular smooth muscle cell differentiation	20p12.2	Cell–Cell Junction/Adhesion	Yes, Alagille syndrome	Yes, reported in isolated cases	Congenital heart disease, possible isolated aortic aneurysm	N/A	AD	No Evidence	D	601920	[20,21,23,24,58]
*KCNN1*	Potassium channel, calcium-activated, intermediate/small conductance, subfamily N, member 1	Voltage-independent calcium-activated potassium channel, contributing to the depolarization of smooth muscle and endothelial cells	19p13.11	Metabolic and Ion Transport	No	Yes (based on proposed mechanistic link to SMC contractility and adaptation)	Vascular tone impairment, arterial stiffness, possible aneurysm predisposition	N/A	N/A	N/A	N/A	602982	[20,21,23,24,59]
*LMNA*	Lamin A/C	Structural nuclear envelope protein; involved in mechanotransduction	1q22	Smooth Muscle Contraction/Mechanotransduction	No	Yes	Aortic aneurysm associated with cardiomyopathy and conduction defects	N/A	AD	N/A	N/A	150330	[20,21,23,24,60]
*LOX*	Lysyl oxidase	Initiates crosslinking of collagens and elastin	5q23.1	Extracellular and Structural Matrix	No	Yes—FTAAD type 10	Aortic aneurysm, dissection, elastic lamellae defects, variable penetrance	Standard	AD	Strong	A2	153455	[20,21,23,24,61,62]
*LTBP1*	Latent transforming growth factor beta-binding protein 1	Regulates the availability of TGF-β ligands in the extracellular matrix	2p22.3	TGF-β/SMAD Signaling	Proposed	No	Connective tissue phenotype with possible vascular features	N/A	AR	N/A	N/A	150390	[20,21,23,24,63]
*LTBP2*	Latent transforming growth factor-beta-binding protein 2	Anchors latent growth factor complexes to microfibrils	14q24.3	Extracellular and Structural Matrix	No clear evidence	No	Ocular phenotypes, glaucoma-related connective tissue dysfunction	N/A	AR	N/A	N/A	602091	[20,21,23,24,63]
*LTBP3*	Latent transforming growth factor beta-binding protein 3	Binds TGF-β and modulates ECM and TGF-β bioavailability	11q13.1	TGF-β/SMAD Signaling	Yes—described in a syndromic context with aortic dilation	No	Aortic aneurysm (often syndromic), variable skeletal/connective features	N/A	AR	N/A	N/A	602090	[20,21,23,24,63]
*LUM*	Lumican	Small leucine-rich proteoglycan	12q21.33	Overlap/Dual-Phenotype/Emerging Candidates	No	Possibly	Aortic wall fragility and aneurysm susceptibility due to impaired ECM assembly	N/A	N/A	N/A	N/A	600616	[20,21,23,24,64]
*MAT2A*	Methionine adenosyltransferase II, alpha	Extrahepatic methionine adenosyltransferase, adds methyl groups for DNA, RNA, protein, and lipid modifications—specifically DNA methylation for epigenetic expressivity of vascular gene expression	2p11.2	Metabolic and Ion Transport	No	Yes	Aberrant methylation profiles; possible smooth muscle dysfunction and matrix turnover abnormalities	Standard	AD	N/A	N/A	601468	[20,21,23,24,65]
*MED12*	Mediator complex subunit 12	Bridges transcription factors to RNA polymerase II, influencing vascular SMC function	Xq13.1	Transcriptional/Nuclear Envelope Regulation	Yes, in the context of Hardikar syndrome	No	Impaired SMC contractility and matrix remodeling; rare X-linked disorder with cardiac anomalies	N/A	X-linked	N/A	N/A	300188	[20,21,23,24,66]
*MFAP5*	Microfibrillar-associated protein 5	Integrates microfibrils into regular elastic recoil	12p13.31	Extracellular and Structural Matrix	No	Yes	Familial thoracic aortic aneurysm and dissection due to impaired microfibril integrity and TGF-β dysregulation	Standard	AD	Moderate	Uncertain	601103	[20,21,23,24,67]
*MYH11*	Smooth muscle myosin heavy chain 11	Major contractile protein in vascular smooth muscle	16p13.11	Smooth Muscle Contraction/Mechanotransduction	No	Yes	TAAD with or without PDA (patent ductus arteriosus), CAD, carotid IA	4.5–5.0 cm	AD	Definitive	A2	160745	[20,21,23,24,68,69]
*MYLK*	Myosin light chain kinase	Regulates smooth muscle contraction via phosphorylation of myosin	3q21.1	Smooth Muscle Contraction/Mechanotransduction	No	Yes	Aortic dissection (with minimal dilation), intrafamilial variability	4.5–5.0 cm	AD	Definitive	A2	600922	[20,21,23,24,70,71]
*NOTCH1*	Notch receptor 1	Primary receptor for Jagged-1, transcriptional regulator for cell fate of smooth muscle/ECM remodeling	9q34.3	Cell–Cell Junction/Adhesion	No	Yes, associated with BAV and aortic aneurysm	BAV, calcific valve disease, ascending aneurysm	Standard	AD	Limited	B2	190198	[20,21,23,24,72,73]
*PDLIM3*	PDZ and LIM domain protein 3 (a.k.a. ALP)	Actin-associated protein involved in cytoskeletal signaling	4q35.1	Smooth Muscle Contraction/Mechanotransduction	No	Possibly	SCAD, thoracic aortic aneurysm/dissection (rare), cardiomyopathy	N/A	Likely AD	N/A	N/A	605889	[20,21,23,24,74]
*PKP2*	Plakophilin 2	Core desmosome component, tethering intermediate filaments between adjacent cells to maintain cohesion during cyclic motion	12p11.21	Cell–Cell Junction/Adhesion	No	Possibly	Arrhythmogenic right ventricular cardiomyopathy (ARVC), the most common cause, with fibrofatty myocardial replacement, ventricular arrhythmias, and sudden death; Brugada syndrome; idiopathic ventricular fibrillation; biventricular and left-dominant cardiomyopathy variants	N/A	AD	N/A	N/A	602861	[20,21,23,24,75]
*PLOD1*	Procollagen-Lysine, 2-oxoglutrate 5-dioxygenase 1	Lysyl hydroxylase that forms hydroxylysine in collagens, allowing for later glycosylation/crosslinking	1p36.22	Extracellular and Structural Matrix	Yes—Ehlers–Danlos syndrome, kyphoscoliotic type (EDSKSCL1)	No	Progressive kyphoscoliosis, joint/skin hyperextensibility, hypotonia; indirect vessel involvement	N/A	AR	N/A	N/A	153454	[14,20,21,23,24,76]
*PLOD3*	Procollagen-Lysine, 2-oxoglutrate 5-dioxygenase 3	Lysyl hydroxylase that forms hydroxylysine in collagens, allowing for later glycosylation/crosslinking—additional galactosyltransferase and glucosyltransferase activities	7q22.1	Extracellular and Structural Matrix	Yes—BCARD syndrome with vascular features (connective tissue + arteriopathy)	No	Craniofacial dysmorphism, scoliosis, osteopenia, vascular fragility (BCARD syndrome)	N/A	AR	No Evidence	D	603066	[14,20,21,23,24,76]
*PMEPA1*	Prostate transmembrane protein, androgen induced 1	Transmembrane protein induced by TGF-β, promoting the degradation of activated SMADs	20q13.31	Transcriptional/Nuclear Envelope Regulation	No	No	Modulation of fibrosis and ECM remodeling; role in vascular remodeling is theoretical	N/A	N/A	N/A	N/A	606564	[20,21,23,24,77]
*PPP1R12A*	Protein phosphatase 1, regulatory subunit 12A	Regulatory subunit of myosin light chain phosphatase	12q21.2-q21.31	Smooth Muscle Contraction/Mechanotransduction	No	Proposed disease modifier	SMC relaxation and cytoskeletal remodeling; potential influence on aortic integrity	N/A	AD	N/A	N/A	602021	[20,21,23,24,62]
*PRDM5*	PR domain-containing protein 5	DNA-binding zinc finger transcription factor binds collagen and proteoglycan-expressing regions	4q27	Transcriptional/Nuclear Envelope Regulation	Yes, brittle cornea syndrome 2	No	ECM disruption in BCS2, corneal fragility, possible vascular contributions	N/A	AR	N/A	N/A	614161	[20,21,23,24,78]
*PRKG1*	cGMP-dependent protein kinase 1	Regulates smooth muscle contraction via calcium desensitization	10q11.23	Smooth Muscle Contraction/Mechanotransduction	No	Yes	Familial aortic dissection with minimal dilation; early-onset dissection	4.5–5.0 cm	AD	Definitive	A2	176894	[8,20,21,23,24,79]
*ROBO4*	Roundabout guidance receptor 4	Endothelial receptor that suppresses pathological angiogenesis and vascular permeability via stabilization of the endothelial layer	11q24.2	Extracellular and Structural Matrix	No	Yes	Aortic valve disease, thoracic aneurysm, endothelial barrier dysfunction	Standard	AD	N/A	N/A	607528	[20,21,23,24,80]
*ROCK2*	Rho-associated coiled coil containing protein kinase 2 (ROCK2)	Serine/threonine kinase involved in actin cytoskeleton dynamics	2p25.1	Smooth Muscle Contraction/Mechanotransduction	No	Proposed disease modifier	SMC tone dysregulation, ECM remodeling; proposed role in medial degeneration	N/A	AD	N/A	N/A	604002	[20,21,23,24,81]
*SECISBP2*	Secis-binding protein 2	Selenocysteine insertion sequence-binding protein that leads to synthesis of selenoproteins that protect cells from oxidative damage	9q22.2	Transcriptional/Nuclear Envelope Regulation	No	No	Not directly TAAD-associated; hypothesized to impact ECM via redox imbalance and inflammation	N/A	AR	N/A	N/A	607693	[20,21,23,24,82]
*SELENOT*	Selenoprotein T	Participates in redox homeostasis in mitochondria	3q25.1	Overlap/Dual-Phenotype/Emerging Candidates	No	Possibly	glucose intolerance due to impaired insulin response	N/A	N/A	N/A	N/A	607912	[20,21,23,24,82]
*SKI*	SKI proto-oncogene	Transcriptional repressor in the TGF-β signaling pathway	1p36.33	TGF-β/SMAD Signaling	Yes—associated with Shprintzen–Goldberg syndrome	No	Aneurysms, tortuosity, craniofacial, and skeletal anomalies	Standard	AD	Limited	B	164780	[20,21,23,24,83]
*SLC2A10 (GLUT10)*	Solute carrier family 2 (facilitated glucose transporter), member 10	Facilitative glucose transporter family protein, which plays a role in the intracellular redox environment	20q13.12	Extracellular and Structural Matrix	Yes—arterial tortuosity syndrome	No	Arterial tortuosity, aneurysm, oxidative stress	Standard	AR	Limited	B2	606145	[20,21,23,24,84]
*SLC39A13 (ZIP13)*	Solute carrier family 39 (zinc transporter), member 13 (also known as glucose transporter 10)	Transports zinc, regulating intracellular levels, which affects lysyl hydroxylase activity/GAG chain elongation	11p11.2	Extracellular and Structural Matrix	No	No	Congenital disorder of glycosylation with systemic findings (no clear TAAD link)	N/A	AR	N/A	N/A	608735	[20,21,23,24,85]
*SMAD2*	SMAD family member 2	Intracellular mediator of TGF-β signaling; regulates gene transcription	18q21.1	TGF-β/SMAD Signaling	Yes—associated with connective tissue disorder, Loeys–Dietz syndrome type 6	No	Arterial aneurysms and dissections (ascending, vertebral, carotid)	Standard	AD	N/A	N/A	601366	[11,20,21,23,24]
*SMAD3*	SMAD family member 3	Intracellular mediator of TGF-β signaling; regulates gene transcription	15q22.33	TGF-β/SMAD Signaling	Yes—Loeys–Dietz syndrome type 3	Yes	Arterial aneurysms and dissections, tortuosity, skeletal findings (osteoarthritis)	4.0–4.2 cm	AD	Definitive	A1	603109	[13,20,21,23,24]
*SMAD6*	SMAD family member 6	Inhibitory SMAD that dampens TGF-β and BMP signaling	15q22.31	TGF-β/SMAD Signaling	No	Yes—associated with BAV/TAAD overlap	Aortic valve disease, BAV-associated aortic aneurysm	Standard	AD	No Evidence	D	602931	[20,21,23,24,86]
*SPARC*	Secreted protein, acidic, cysteine-rich (osteonectin)	Influences the synthesis and interaction of proteoglycans and collagens	5q33.1	Extracellular and Structural Matrix	No	No	Associated with Osteogenesis Imperfecta XVII; no clear vascular phenotype	N/A	AR	N/A	N/A		[20,21,23,24,87]
*TBX20*	T-box transcription factor 20	Binds the DNA-binding domain T-box 20, influences cardiomorphogenesis	7p14.2	Transcriptional/Nuclear Envelope Regulation	No	Yes	Associated with BAV and aortic dilation through disrupted valvulogenesis and aortic patterning	N/A	AD	N/A	N/A	606061	[20,21,23,24,88,89]
*TGFB2*	Transforming growth factor beta-2	Ligand in the TGF-β pathway; regulates vascular development and remodeling	1q41	TGF-β/SMAD Signaling	Yes—Loeys–Dietz syndrome type 4	Yes	Aneurysms, arterial tortuosity, other vascular anomalies, BAV	4.5–5.0 cm	AD	Definitive	A1	190220	[20,21,23,24,90]
*TGFB3*	Transforming growth factor beta-3	TGF-β family cytokine; modulates ECM and cellular proliferation	14q24.3	TGF-β/SMAD Signaling	Yes—Loeys–Dietz syndrome type 5	No	Aneurysms, dissections, intracranial aneurysms	Standard	AD	N/A	N/A	190230	[20,21,23,24,91]
*TGFBR1*	Transforming growth factor-beta receptor type I	Serine/threonine kinase receptor for TGF-β; initiates SMAD signaling cascade	9q22.33	TGF-β/SMAD Signaling	Yes—Loeys–Dietz syndrome type 1	Rare, but possible	Aneurysms, arterial tortuosity, craniofacial features (bifid uvula, hypertelorism), skeletal dysplasia	4.0–4.5 cm	AD	Definitive	A1	190181	[11,12,20,21,23,24]
*TGFBR2*	Transforming growth factor-beta receptor type II	Serine/threonine kinase that binds TGF-β ligands and activates TGFBR1	3p24.1	TGF-β/SMAD Signaling	Yes—Loeys–Dietz syndrome type 2	Rare, but possible	Aneurysms, widespread arterial dissection, craniofacial and skeletal features	4.0–4.5 cm	AD	Definitive	A1	190182	[11,12,20,21,23,24]
*THBS2*	Thrombospondin-2	Suppresses metalloproteinase activity, limiting extracellular matrix degradation	6q27	Extracellular and Structural Matrix	Yes—classic-like EDS	No	ECM fragility, connective tissue disease with vascular involvement suspected	N/A	AD	N/A	N/A	188061	[20,21,23,24,92]
*THSD4 (ADAMTSL6)*	Thrombospondin type 1 domain-containing protein 4	Promotes the assembly of microfibrils through fibrillin-1 assembly and suppression of TGF-B	15q23	Extracellular and Structural Matrix	Yes	Yes—familial thoracic aortic aneurysm with variable expressivity	Aortic dilation, familial TAAD (AD pattern with variable penetrance)	N/A	AD	N/A	N/A	614476	[20,21,23,24,93]
*TLN-1*	Talin-1	Cytoskeletal adaptor protein that links integrins to the actin cytoskeleton	9p13.3	Smooth Muscle Contraction/Mechanotransduction	No	Yes	SCAD, thoracic aortic aneurysm, arrhythmogenic cardiomyopathy features (in some cases)	N/A	AD	N/A	N/A	186745	[20,21,23,24,26,94,95]
*TNXB*	Tenascin-XB	Collagen-organizing glycoprotein, which modulates fibril spacing via intercalation	6p21.33-p21.32	Extracellular and Structural Matrix	Yes—EDS classic-like (EDSCLL)	No	Skin hyperextensibility, easy bruising, connective tissue fragility	N/A	AR	N/A	N/A	600985	[20,21,23,24,89]
*VCL*	Vinculin	Actin-binding protein is important in cell–matrix adhesion and mechanical force transmission	10q22.2	Smooth Muscle Contraction/Mechanotransduction	No	Yes	Thoracic aortic aneurysm (rare), SCAD, dilated/arrhythmogenic cardiomyopathy	N/A	AD	N/A	N/A	193065	[20,21,23,24,96]
*ZBTB20*	Zinc finger and BTB-containing protein 20	Influences PI3K and MAPK signaling pathways and has been linked to vascular remodeling and SMC phenotypic modulation	3q13.31	Overlap/Dual-Phenotype/Emerging Candidates	No	Possibly	Disrupted dendritic and synaptic structure; proposed developmental influence	N/A	AD	N/A	N/A	606025	[20,21,23,24,97]
*ZFHX3*	Zinc finger homeobox 3	Zinc finger homeobox gene expressed in vascular tissues and may regulate cytoskeletal gene expression and inflammatory pathways relevant to medial degeneration	16q22.2-q22.3	Overlap/Dual-Phenotype/Emerging Candidates	Possibly	No	Syndromic intellectual disability with vascular anomalies	N/A	AD	N/A	N/A	104155	[20,21,23,24,98]
*ZNF469*	Zinc finger protein 469	Zinc finger transcription factor that regulates ECM gene expression, including collagen and proteoglycans	16q24.2	Overlap/Dual-Phenotype/Emerging Candidates	No	Possibly	ZNF469 participates in extracellular matrix regulation. It downregulates several collagen and adhesion molecules. While it has no confirmed clinical role in TAAD, it may function as a modifier gene. It is known for causing brittle cornea syndrome and could influence vascular fragility due to its ECM effect	N/A	AR	N/A	N/A	612078	[20,21,23,24,78]

Data summarized from the published literature as cited. Table design and synthesis by the authors. Summary: This table catalogs genes associated with heritable thoracic aortic aneurysm and dissection (TAAD), classifying them by protein function, genomic location, and pathway association. Each gene is annotated for involvement in syndromic TAAD (sTAAD), nonsyndromic familial TAAD (nsFTAAD), and linked clinical phenotypes. Additional annotations include aortic size thresholds for surgical intervention, mode of inheritance, strength of evidence, and clinical classification (A–D). Pathway assignments include structural, signaling, metabolic, transcriptional, and mechanosensory categories, reflecting the complexity and pleiotropy of TAAD genetics. All sources are listed in the last column of each row. Abbreviations used in the table include autosomal recessive (AR), autosomal dominant (AD), X-linked (XL), thoracic aortic aneurysm and dissection (TAAD), nonsyndromic familial thoracic aortic aneurysm and dissection (nsFTAAD), extracellular matrix (ECM), aortic aneurysm (AA), smooth muscle cell (SMC), surgical intervention (SI), mode of inheritance (MoA), and strength of association (SoA).

**Figure 2 medsci-13-00155-f002:**
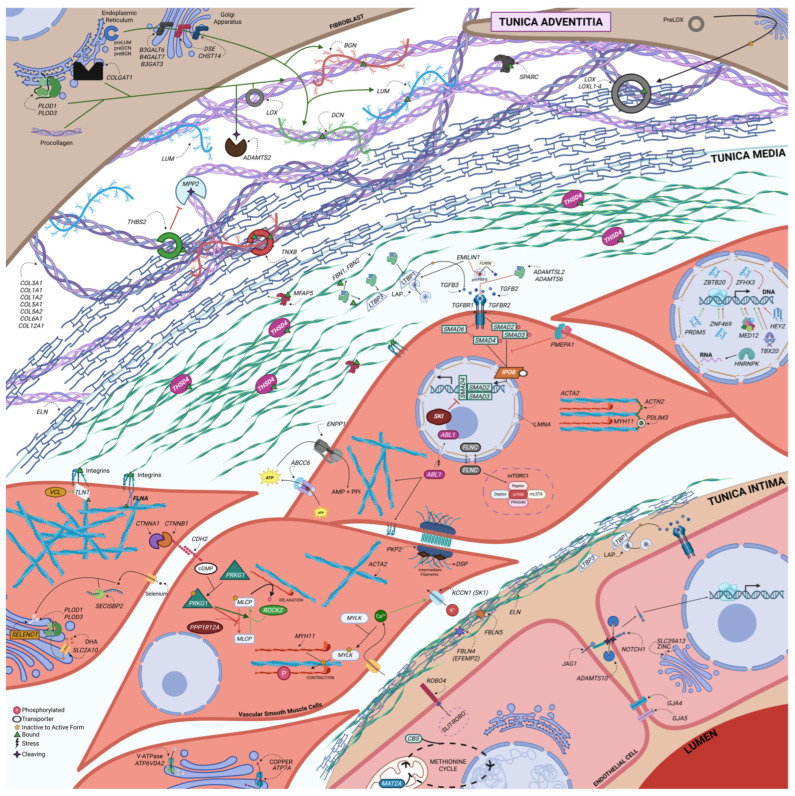
Central illustration: Integrated molecular pathways underlying thoracic aortic aneurysm and dissection. Summary: This illustration maps the major functional pathways implicated in TAAD across the structural layers of the aortic wall, including the tunica intima, media, and adventitia, as well as relevant cell types such as vascular smooth muscle cells, fibroblasts, and endothelial cells. Genes are grouped into mechanistic categories, including TGF-β signaling, extracellular matrix synthesis and crosslinking, cytoskeletal and contractile regulation, cell–cell and cell–matrix adhesion, nuclear transcription and chromatin remodeling, metabolic/redox control, and ion transport. Arrows and symbols denote molecular interactions such as phosphorylation, transport, cleavage, and stress response [99] For complete gene names and classifications by pathway, refer to Figure 3.

**Figure 3 medsci-13-00155-f003:**
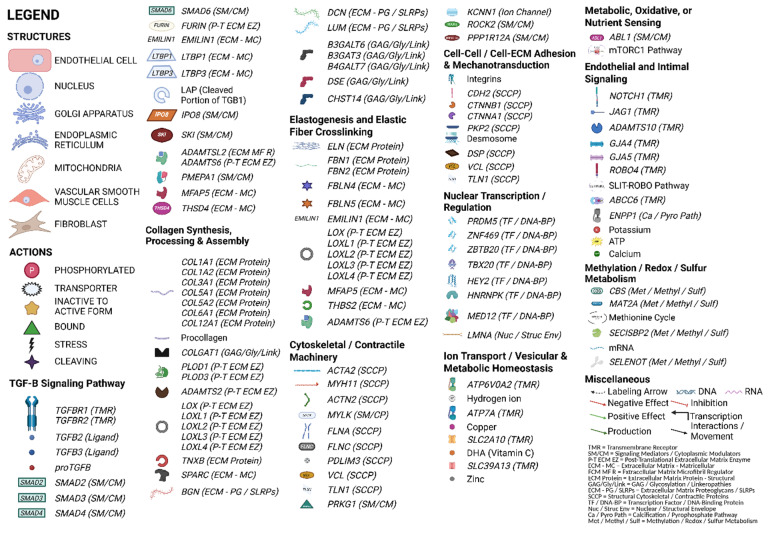
This extended legend provides gene-level detail for the functional pathways depicted in the central illustration. Genes are categorized by their primary biological roles, including TGF-β signaling, extracellular matrix (ECM) organization, cytoskeletal and contractile machinery, nuclear transcriptional regulation, cell–cell and cell–ECM adhesion, proteoglycan biosynthesis, ion transport, metabolism, and redox regulation. Structural components (e.g., smooth muscle cells, fibroblasts, endothelial cells, nuclei, Golgi) and molecular actions (e.g., phosphorylation, transport, cleavage) are annotated using standardized symbols. Each gene is linked to its associated function, protein class, and pathway abbreviation for ease of reference [100]. This figure is intended to complement Figure 2, allowing readers to interpret spatial localization and functional grouping of TAAD-related genes at a glance.

#### 3.1.2. Ligand and Receptor Variants

The canonical TGF-β signaling cascade begins with the binding of dimeric TGF-β ligands to a heteromeric receptor complex composed of type II and type I serine/threonine kinase receptors. These receptors, encoded by *TGFBR2* and *TGFBR1*, respectively, phosphorylate receptor-regulated SMADs upon activation. Pathogenic variants in *TGFBR1* and *TGFBR2* are the defining causes of Loeys–Dietz syndrome (LDS) types I and II and lead to widespread arteriopathy, including early-onset dissections at smaller aortic diameters [11,12]. The vascular phenotype is often accompanied by systemic features such as craniosynostosis, bifid uvula, and skeletal dysplasia. These receptor variants, typically heterozygous and often de novo, impair canonical signaling yet paradoxically result in increased downstream SMAD phosphorylation in affected tissues [11,101], suggesting compensatory or alternative pathway activation.

Genetic variation in upstream ligands also contributes to LDS via altered ligand dosage and receptor engagement. *TGFB2* and *TGFB3*, which encode TGF-β2 and TGF-β3 ligands, cause LDS types IV and V, respectively. *TGFB2* variants result in a phenotype overlapping with Marfan syndrome and are associated with thoracic aneurysms and mitral valve prolapse [90], while *TGFB3* variants present with craniofacial anomalies, joint laxity, and aortic dilation [91]. Despite representing distinct subtypes, both genes converge on impaired ligand availability and disrupted ECM regulation, reinforcing the importance of precise TGF-β ligand titration in aortic wall integrity.

#### 3.1.3. Intracellular Mediators and Transcriptional Regulation

Intracellularly, receptor-mediated signaling proceeds through SMAD transcription factors. *SMAD3*, which encodes a receptor-activated SMAD family member 3, is associated with LDS type III, also referred to as AOS. Patients exhibit aggressive vascular disease alongside early-onset osteoarthritis [13]. *SMAD3* variants are among the most penetrant in the TGF-β pathway and are associated with dissection at modest aortic diameters. *SMAD2*, a closely related gene, has been implicated through rare damaging variants, though genotype–phenotype correlations remain less well-defined [11]. The balance of nuclear SMAD activity is also influenced by repressors; *SKI*, a transcriptional inhibitor of SMAD-mediated gene expression, is associated with SGS. Heterozygous loss-of-function *SKI* variants result in disrupted repression of TGF-β target genes, contributing to craniosynostosis, connective tissue fragility, and aortic aneurysm [83].

#### 3.1.4. Negative Feedback and Nuclear Transport

Negative feedback within the TGF-β axis is further regulated by inhibitory SMADs and nuclear transport machinery. *SMAD6* encodes an inhibitory SMAD that dampens both TGF-β and bone morphogenic protein (BMP) signaling. Although not classically associated with syndromic TAAD, *SMAD6* variants are observed in patients with bicuspid aortic valve (BAV) and associated aortopathies, where they may enhance SMC remodeling under abnormal hemodynamics [86]. Additionally, importin-8 (encoded by *IPO8)* is required for nuclear translocation of SMADs following receptor activation. Biallelic loss-of-function variants in *IPO8* impair this process, resulting in a syndromic connective tissue disorder (VISS syndrome) with cardiovascular, skeletal, and immune abnormalities [56,57]. Together, *SMAD6, SKI,* and *IPO8* illustrate the diverse regulatory checkpoints that control nuclear SMAD signaling fidelity.

#### 3.1.5. Extracellular Matrix Modulators

Beyond the ligand–receptor–SMAD core, the bioavailability and spatial regulation of TGF-β signaling are modulated by extracellular matrix tethering proteins. LTBP1 and LTBP3 (latent TGF-β binding proteins 1 and 3) anchor TGF-β complexes within the ECM, governing their localized activation in response to mechanical stress or enzymatic cleavage [63]. Disruption of this mechanism alters TGF-β gradient formation and signaling kinetics. Notably, preliminary reports have also linked heterozygous variants in *LTBP3* to thoracic aortic aneurysm and dissection in autosomal dominant contexts, suggesting that *LTBP3* may play a more direct and clinically relevant role in aortic pathology than previously appreciated. Emilin 1 also participates in this regulatory layer by facilitating TGF-β precursor maturation and contributing to elastic fiber structure. Though human pathogenic variants have not been established, *EMILIN1*-deficient mouse models exhibit aortic root aneurysms and elevated TGF-β activity, suggesting a modifying role in vascular disease [45].

#### 3.1.6. Paradoxical Hyperactivation and Therapeutic Implications

Notably, genetic variation across this signaling axis, whether it impairs receptor function, reduces ligand levels, or interferes with SMAD regulation, consistently results in paradoxical hyperactivation of downstream signaling. Elevated phosphorylated SMAD2 (pSMAD2) levels in aortic tissue are common even in loss-of-function contexts, likely due to defective negative feedback, receptor recycling, or compensatory activation of parallel cascades (e.g., MAPK/ERK). This paradox has major therapeutic implications, including interest in pharmacologic TGF-β pathway modulators, such as angiotensin receptor blockers (ARBs), which may attenuate maladaptive ECM remodeling.

#### 3.1.7. Clinical Significance

In sum, the TGF-β signaling pathway represents a highly conserved, multi-tiered regulatory system that governs vascular structure through ECM synthesis, SMC phenotype, and inflammatory balance. Disruption at any level, from ligand production to nuclear SMAD translocation, can precipitate syndromic or nonsyndromic aortopathy. The inclusion of TGF-β pathway genes on clinical panels is supported by their strong penetrance and clinical utility in guiding early surgical intervention, with the 2022 AHA/ACC guidelines recommending prophylactic repair at 4.0–4.5 cm in individuals with *TGFBR1, TGFBR2, SMAD3, TGFB2,* or *TGFB3* variants [19,102].

### 3.2. Smooth Muscle Cell Contractility and Mechanotransduction

#### 3.2.1. Core Mechanotransduction Overview

The mechanical integrity of the thoracic aorta depends on the ability of VSMCs to sense and respond to hemodynamic stress through contractile force generation and cytoskeletal anchoring. This dynamic regulation, known as mechanotransduction, requires a complex network of contractile proteins, signaling kinases, cytoskeletal linkers, and force-transducing junctions. Genetic disruptions within this system can lead to VSMC phenotypic modulation, ECM disarray, and aortic wall weakening, all of which are central to the pathogenesis of TAAD (refer to Table 1, Figure 2).

#### 3.2.2. Actomyosin Machinery

At the core of SMC contraction is the actomyosin machinery, which generates force through cyclic interactions between actin filaments and myosin motor proteins. Smooth muscle α-actin, encoded by *ACTA2,* is a central structural protein in the thin filament complex. Variants in *ACTA2* are the most common cause of nonsyndromic familial TAAD and result in impaired actin polymerization, leading to reduced contractile capacity and early-onset dissection [25,31]. Smooth muscle myosin heavy chain 11 (encoded by *MYH11)* forms the thick filament and drives contractile force via ATP-dependent cross-bridge cycling with actin. Pathogenic variants in *MYH11* disrupt this interaction and are associated with familial TAAD, often accompanied by patent ductus arteriosus (PDA) [68,69]. Together, ACTA2 and MYH11 form the fundamental sliding filament system that powers vascular tone and wall support.

#### 3.2.3. Kinase Regulation of Contractility

The initiation of SMC contraction depends on calcium-mediated signaling, particularly through the phosphorylation of myosin light chain (MLC). Myosin light chain kinase (encoded by *MYLK*) is activated by calcium–calmodulin and directly phosphorylates MLC to trigger contraction. Loss-of-function *MYLK* variants reduce MLC phosphorylation, compromising contractility and increasing susceptibility to aortic dissection, even at small diameters [70,71]. In contrast, cGMP-dependent protein kinase I (encoded by *PRKG1*) exerts an opposing vasodilatory influence by promoting MLC dephosphorylation and SMC relaxation. Pathogenic *PRKG1* variants cause a gain-of-function phenotype that leads to excessive vasorelaxation, decreased wall tension, and progressive medial degeneration [8,79]. These opposing kinases illustrate how dysregulated signal transduction, not just structural protein defects, can destabilize the aortic wall.

#### 3.2.4. Cytoskeletal–Nuclear and ECM Linkers

The contractile apparatus must be tightly linked to both the cytoskeleton and extracellular matrix to transmit force effectively. This is accomplished by a network of scaffold and linker proteins that integrate mechanical signals from focal adhesions and adherens junctions. Filamin A (encoded by *FLNA*) is an actin-binding protein that crosslinks actin filaments and serves as a mechanosensitive hub for signaling complexes at the cell cortex. Pathogenic variants in *FLNA* are associated with X-linked periventricular nodular heterotopia and vascular anomalies, including aortic aneurysms, highlighting their dual role in cytoskeletal organization and vascular development [50]. Nuclear envelope proteins Lamin A and C (encoded by *LMNA*) connect the cytoskeleton to the nuclear interior via the LINC (linker of nucleoskeleton and cytoskeleton) complex, enabling mechanical signals to influence gene expression. Although *LMNA* is more commonly associated with cardiomyopathy, muscular dystrophy, and progeroid syndromes, its role in mechanotransduction and nuclear stiffness suggests a contributory role in aortic disease as well. In addition to *FLNA* and *FLNC*, filamin B (encoded by *FLNB*) may also contribute to vascular wall integrity. Although its role in thoracic aortic disease is less well-established, filamin B shares structural and functional similarities with other filamins and has been implicated in vascular development and homeostasis, warranting further study [20,60].

#### 3.2.5. Focal Adhesion Proteins

Emerging data suggest that additional focal adhesion proteins and mechanosensory elements contribute to SMC function and aortic wall remodeling. Talin-1 (encoded by *TLN1*) anchors integrins to the actin cytoskeleton and is essential for force transmission at focal adhesions. Downregulation of TLN1 has been observed in dissected aortic tissue and is associated with increased VSMC proliferation and migration, indicating a shift toward a synthetic phenotype that may accelerate medial degeneration [26,94]. Vinculin (encoded by *VCL*), a talin-binding partner, reinforces focal adhesions under stress and may similarly affect cellular contractility and survival [96]. PDZ-LIM domain protein 3 (encoded by *PDLIM3*) helps organize actin filaments and transduce force within the contractile apparatus, although its genetic contribution to TAAD remains under investigation [74].

#### 3.2.6. RhoA-ROCK Signaling Axis

Regulatory components of the RhoA-ROCK signaling axis are also relevant to aortic wall mechanics. Rho-associated coiled coil containing protein kinase 2 (encoded by *ROCK2*) promotes actomyosin contractility by phosphorylating MLC and inhibiting myosin phosphatase, thereby sustaining contraction [81]. Protein phosphatase 1, regulatory subunit 12A (encoded by *PPP1R12A*), a myosin phosphatase-targeting subunit, counterbalances ROCK activity by facilitating MLC dephosphorylation and SMC relaxation. While not yet confirmed as monogenic TAAD genes, dysregulation of this axis can influence SMC tone, ECM secretion, and response to biomechanical stress, potentially acting as disease modifiers [62].

#### 3.2.7. Sarcomere-Associated Genes

Lastly, a set of sarcomere-associated genes traditionally linked to cardiomyopathy has gained attention for possible roles in vascular disease. *ACTN2* encodes α-actinin-2, which crosslinks actin filaments at the Z-disc and modulates myofibrillar tension. Rare *ACTN2* variants have been observed in patients with left-dominant arrhythmogenic cardiomyopathy and vascular complications [103]. Filamin C (encoded by *FLNC*), closely related to FLNA, plays similar roles in actin crosslinking in muscle cells. Though best known for skeletal and cardiac myopathies, the altered expression of Filamin C in aortic tissue suggests that it may also influence cytoskeletal integrity in VSMCs [51].

#### 3.2.8. Summary and Clinical Implications

Together, these genes define a highly integrated network of proteins that link cytoskeletal structure, contractile force generation, and mechanosensitive signaling. Disruption at any level, whether in actomyosin machinery, signaling kinases, focal adhesion components, or nuclear–cytoskeletal connectors, can compromise aortic wall stability, promote maladaptive remodeling, and increase the risk of dissection. Continued investigation into how these elements interact under mechanical load will be essential for advancing precision diagnostics and targeted therapies in genetically driven TAAD.

### 3.3. Extracellular Matrix and Structural Matrix Genes

#### 3.3.1. Overview of ECM Function in TAAD

The ECM of the thoracic aorta provides the fundamental scaffold that supports tensile strength, elasticity, and dynamic responsiveness to hemodynamic stress. Its proper function depends on the coordinated synthesis, assembly, crosslinking, and remodeling of elastic fibers, collagen fibrils, and proteoglycans. Genetic disruption of these interlinked components weakens the aortic wall and predisposes to aneurysm and dissection (refer to Table 1, Figure 2). Below, we highlight key ECM-related genes, emphasizing their molecular interplay within shared functional networks.

#### 3.3.2. Elastic Fiber Network

The elastic fiber network forms the mechanical foundation of the proximal aorta. Fibrillin-1, encoded by *FBN1*, is the central component of microfibrils that scaffold elastic fiber assembly and sequester latent TGF-β complexes [48]. Fibrillin-2 (encoded by *FBN2*) is structurally similar to fibrillin-1 and is predominantly expressed in fetal development, where it participates in early microfibril formation [16]. Together, fibrillin-1 and fibrillin-2 provide the molecular lattice on which elastin is deposited. Pathogenic variants in *FBN1* underlie Marfan syndrome, while *FBN2* is implicated in congenital contractural arachnodactyly and may modify risk in syndromic or nonsyndromic arteriopathies. Anchoring this scaffold are fibulins, including EGF-containing fibulin-like extracellular matrix protein 2 and fibulin 5 (encoded by *FBLN4 and FBLN5,* respectively), which interact directly with fibrillin-1, elastin, and lysyl oxidases to promote elastic fiber crosslinking and stability [20,47]. Loss-of-function variants in *FBLN4* cause recessive cutis laxa with severe vascular involvement, whereas *FBLN5* variants lead to elastic fiber fragmentation. Microfibrillar-associated protein 5 (encoded by *MFAP5*), another fibrillin-binding protein, integrates with microfibrils to regulate elastic recoil and fine-tune TGF-β signaling during ECM remodeling [20,67]. At the core of the elastic fiber itself is elastin, encoded by *ELN*, whose polymerization and crosslinking are essential for vessel wall compliance. *ELN* haploinsufficiency, as seen in Williams syndrome, results in supravalvular aortic stenosis and proximal arteriopathy [44].

#### 3.3.3. Collagen Biosynthesis and Organization

Parallel to elastic fiber dynamics, collagen biosynthesis and fibril organization provide longitudinal tensile strength to the aorta. Type III collagen, encoded by *COL3A1*, is the dominant fibrillar collagen in the medial layer and is deficient in vascular Ehlers–Danlos syndrome (vEDS), predisposing to spontaneous rupture [14]. *COL1A1* and *COL1A2*, forming type I collagen, are foundational to the interstitial matrix, while *COL5A1* and *COL5A2*, associated with classical EDS, form type V collagen, which regulates heterotypic fibrillogenesis and modulates fibril diameter [14,37]. *COL12A1* encodes for a FACIT (fibril-associated collagen with interrupted triple helices) collagen, which tethers fibrils to surrounding structures, enhancing mechanical cohesion [36]. Fibrillar stability is further reinforced by the protein Collagen Beta (1-O) Galactosyltransferase 1 encoded by *COLGALT1*, which glycosylates hydroxylysine residues and stabilizes triple-helix formation [38].

#### 3.3.4. Post-translational Crosslinking Enzymes

This fibrillar collagen framework is dynamically modified by post-translational enzymes, which collectively govern ECM stiffness, crosslinking, and turnover. Lysyl oxidase (encoded by *LOX*) and its paralogs lysyl oxidase-like 1–4 catalyze lysine-derived aldehyde formation in both collagen and elastin, enabling covalent crosslinks. *LOX* variants are associated with familial TAAD and impair ECM tensile strength [20,61,62,94]. *PLOD1* and *PLOD3* encode lysyl hydroxylases, which hydroxylate lysine residues in collagen and other collagen-like molecules, a critical precursor step to glycosylation and crosslinking [14,76]. Notably, the protein encoded by *PLOD3* also has galactosyltransferase and glucosyltransferase activities, making it a dual-function ECM modifier [76]. *ADAMTS2* encodes a procollagen N-proteinase, which removes propeptides to allow proper fibril self-assembly, with biallelic mutations resulting in dermatosparaxis EDS [27]. These enzymes act in a coordinated sequence, wherein PLOD-mediated hydroxylation facilitates COLGALT1 glycosylation and LOX-driven crosslinking, highlighting the interdependence of collagen maturation pathways.

#### 3.3.5. Organizational Glycoproteins

Beyond fibrillar assembly, the ECM relies on organizational proteins and glycoproteins that regulate spatial orientation, mechanical resilience, and cell–matrix signaling. *TNXB*, encoding tenascin-X, is a collagen-organizing glycoprotein that intercalates into fibril networks and modulates fibril spacing and integration. While many vascular complications in EDS are associated with *COL3A1* variants, tenascin-X deficiency in classical-like EDS may also predispose to arterial fragility by impairing collagen bundle cohesion [89]. *THBS2*, encoding thrombospondin-2, functions as a matricellular protein that suppresses metalloproteinase activity, thereby limiting ECM degradation. Its absence leads to excessive matrix turnover and dysregulated remodeling, which may underlie increased susceptibility to aneurysm formation [92].

#### 3.3.6. ECM-TGF-β Modulators

Several regulatory ECM proteins modulate growth factor bioavailability and signal transduction, particularly within the TGF-β pathway. *LTBP1*, *LTBP2*, and *LTBP3* encode latent TGF-β binding proteins, which anchor latent complexes to microfibrils, coordinating spatial and temporal activation [63]. *EMILIN1* encodes for an elastic fiber-associated glycoprotein that regulates TGF-β precursor maturation and has been shown in mouse models to cause aortic root aneurysms when disrupted, likely via dysregulated signaling [45]. Thrombospondin type 1 domain-containing protein 4 (encoded by *THSD4 (ADAMTSL6)*) facilitates fibrillin-1 microfibril assembly and suppresses pathological TGF-β activation and is associated with both syndromic and nonsyndromic TAAD [93].

#### 3.3.7. Proteoglycan and GAG Metabolism

The proteoglycan and glycosaminoglycan (GAG) matrix, which maintains hydration and buffering capacity, is orchestrated by enzymes that form linker regions and assemble GAG chains. *B3GALT6* and *B4GALT7* encode beta-1,3-galactosyltransferase 6 and beta-1,4-galactosyltransferase 7, which help catalyze early steps in the GAG–core protein linkage region; variants in either gene cause spondylodysplastic EDS with vascular manifestations [30,31]. *DSE* and *CHST14* encode dermatan sulfate epimerase and carbohydrate sulfotransferase 14, respectively. Variants in *DSE* and *CHST14* are mutated in musculocontractural EDS and contribute to ECM softening and arterial tortuosity [35,42]. Osteonectin (encoded by *SPARC*) and microfibrillar-associated protein 5 (encoded by *MFAP5*), though structurally unrelated, both influence proteoglycan–collagen interactions and may affect vessel wall hydration and compliance through modulation of fibrillar anchoring [21,67,87].

#### 3.3.8. Golgi/Ion Transport in ECM Processing

Further integration occurs at the level of Golgi-localized and ion transport pathways required for ECM post-processing. ATPase, H+ Transporting, Lysosomal, V0 Subunit A2 (encoded by *ATP6V0A2)*, a subunit of the vacuolar ATPase, maintains Golgi acidification required for proper glycosylation of matrix proteins; its loss causes cutis laxa with tortuous and aneurysmal arteries [28]. Solute carrier family 2 (facilitated glucose transporter), member 10 (encoded by *SLC2A10*), and solute carrier family 39 (zinc transporter), member 13 (encoded by *SLC39A13 (ZIP13)*), regulate intracellular redox and zinc levels, respectively, which in turn affect lysyl hydroxylase activity and GAG chain elongation [84,85].

#### 3.3.9. Additional ECM-Interacting Proteins

Finally, endothelial–matrix interface proteins encoded by genes, such as *ROBO4,* mediate vascular homeostasis. *ROBO4* encodes an endothelial receptor that suppresses pathological angiogenesis and vascular permeability. Its disruption destabilizes the endothelium and may impair communication with underlying ECM scaffolds, contributing to arterial wall vulnerability in TAAD [80].

#### 3.3.10. Summary and Clinical Implication

Together, these genes represent a highly interconnected matrix signaling system. Disruptions in fibril formation, crosslinking, proteoglycan synthesis, and matrix–cell communication converge to undermine the structural and signaling integrity of the aortic wall. This reinforces the view that ECM-related arteriopathies arise not from isolated failures but from coordinated breakdowns across a matrix-dependent signaling axis.

### 3.4. Cell–Cell Junctions and Aortic Wall Cohesion

#### 3.4.1. Overview of Cell–Cell Junctions in Aortic Wall Integrity

The structural cohesion of the aortic wall depends not only on the ECM and smooth muscle contractility but also on robust intercellular junctions that maintain mechanical integrity across the vascular media. These junctional complexes, comprising adherens junctions, desmosomes, focal adhesions, and gap junctions, coordinate force transmission, cellular communication, and cytoskeletal organization. Disruption of these systems can destabilize the aortic wall under stress and contribute to aneurysm formation or dissection (refer to Table 1, Figure 2).

#### 3.4.2. Notch Pathway Regulation of VSMC Differentiation

A central component of cell–cell cohesion in VSMCs is adherens junction signaling, particularly the Notch pathway. Jagged1 (encoded by *JAG1*) serves as a membrane-bound Notch ligand that plays a key role in embryonic vascular development and VSMC differentiation. While JAG1 variants are classically associated with Alagille syndrome, isolated cases of thoracic aortic aneurysm have been observed even in the absence of syndromic features, suggesting that impaired Notch signaling may destabilize arterial structure [58]. Notch Receptor 1, the primary receptor for Jagged1, is a transcriptional regulator of SMC fate and ECM remodeling. Variants in *NOTCH1* have been implicated in congenital heart defects and BAV, both of which are associated with ascending aortopathy [73]. Dysregulation of Notch Receptor 1 affects both endothelial and medial compartments of the vessel wall, altering cellular adhesion, ECM production, and differentiation. The interaction between Jagged1 and Notch Receptor 1 thus represents a critical axis for maintaining cellular organization and arterial identity throughout vascular development and homeostasis.

#### 3.4.3. Adherens Junctions

The integrity of adherens junctions and cadherin–catenin complexes is further supported by N-cadherin 2 (encoded by *CDH2*), a calcium-dependent transmembrane adhesion molecule essential for SMC-SMC cohesion. N-cadherin 2 anchors the actin cytoskeleton through catenins, forming stable intercellular contacts that are critical for maintaining VSMC alignment under mechanical load. Experimental data indicate that loss of N-cadherin 2 disrupts cytoskeletal anchoring and promotes VSMC dedifferentiation, a hallmark of aneurysmal degeneration [34]. Although pathogenic variants in *CDH2* have not been firmly linked to monogenic TAAD, its central role in medial wall architecture positions it as a strong functional candidate.

#### 3.4.4. Desmosomal Proteins

Beyond adherens junctions, desmosomal proteins contribute to intracellular force distribution and cell–cell stability. Desmoplakin (encoded by *DSP*) and plakophilin-2 (encoded by *PKP2*) are core components of desmosomes, which tether intermediate filaments across adjacent cells and maintain cohesion during cyclic stretch. While both are best known for their role in arrhythmogenic cardiomyopathies, they are also expressed in VSMCs. Disruption of desmosomal anchoring may reduce medial wall resilience, especially in regions subject to high mechanical stress [43,75]. Their dual role in cardiomyocytes and VSMCs suggests a broader phenotype that can include aortic fragility alongside cardiac conduction abnormalities.

#### 3.4.5. ECM–Cell Anchoring Proteins

ECM-interacting proteins, such as thrombospondins and fibrillin-binding partners, serve to anchor cells to the surrounding matrix, linking adhesion receptors with structural components. *THSD4* (also known as *ADAMTSL6*) encodes thrombospondin type 1 domain-containing protein 4, a secreted ECM-associated protein, which facilitates fibrillin-1 microfibril assembly and suppresses aberrant TGF-β signaling. Germline variants in *THSD4* have been observed in both syndromic and nonsyndromic familial TAAD, and functional studies suggest that THSD4 loss compromises matrix stability and cell–matrix cohesion [93]. Because THSD4 acts at the interface of ECM organization and intracellular signaling, its disruption may weaken both structural scaffolding and dynamic remodeling cues.

#### 3.4.6. Focal Adhesion Mechanosensors

Anchoring these connections to the cytoskeleton, talin-1 (encoded by *TLN1*) plays a pivotal role in focal adhesion complexes by linking integrins to actin filaments. Talin-1 serves as a mechanotransducer that translates extracellular forces into intracellular signaling responses, allowing SMCs to adapt to mechanical strain. Reduced *TLN1* expression in aortic dissection tissue correlates with increased SMC proliferation and migration, indicating a switch from a contractile to a synthetic phenotype and a breakdown in wall stability [26]. Rare missense variants in *TLN1* have also been associated with spontaneous coronary artery dissection, reinforcing its broader role in vascular cohesion and mechanotransduction [95].

#### 3.4.7. Summary and Clinical Implications

Together, these genes illustrate the multilayered architecture of aortic wall cohesion, spanning from Notch-regulated SMC differentiation genes (*JAG1*, *NOTCH1*) to adherens and desmosomal junctions (*CDH2*, *DSP*, *PKP2*) to ECM coupling and mechanosensory anchoring (*THSD4*, *TLN1*). Disruption in any of these systems can compromise intercellular force transmission and wall stability, particularly under pulsatile mechanical load. These mechanisms may be especially relevant in regions of high curvature or stress concentration, where local failure in cell–cell cohesion may function as the initiating site for medial degeneration or dissection.

### 3.5. Transcriptional Regulators and Nuclear Envelope Genes

#### 3.5.1. Overview of Nuclear and Transcriptional Regulation in TAAD

The structural integrity and function of the thoracic aorta are governed not only by ECM architecture and cytoskeletal mechanics but also by transcriptional networks and nuclear envelope components that orchestrate gene expression, chromatin organization, and cellular adaptation to biomechanical stress. Genetic alterations in these regulatory systems can disrupt ECM production, SMC differentiation, and oxidative homeostasis, thereby predisposing to TAAD (refer to Table 1, Figure 1, Figure 2, Figure 3 and Figure 4).

#### 3.5.2. Nuclear Envelope and Mechanotransduction

At the interface between the cytoskeleton and the genome, *LMNA* encodes Lamin A/C, key components of the nuclear lamina that support nuclear structure and transmit mechanical signals from the actin cytoskeleton via the LINC complex. Variants in *LMNA* are associated with a broad spectrum of laminopathies, including cardiomyopathies and progeroid syndromes, which often include vascular features, such as aortic dilation and medial degeneration. These manifestations mirror histopathologic findings in TAAD, such as elastin fragmentation and SMC loss, and suggest a mechanotransductive role for Lamin dysfunction in aortic disease, even though *LMNA* is not yet formally recognized as a monogenic TAAD gene [60].

#### 3.5.3. Transcription Factors in Aortic Wall Remodeling

Several transcriptional regulators implicated in TAAD converge on pathways controlling ECM synthesis and vascular smooth muscle phenotype. *PRDM5* encodes a zinc finger transcription factor that regulates the expression of collagens and proteoglycans and is best known for its role in brittle cornea syndrome. However, its influence on ECM organization extends to the vasculature, where dysregulation may compromise the structural stability of the arterial wall [95]. HEY2, a basic helix–loop–helix transcription factor and downstream effector of Notch signaling, regulates SMC differentiation and endothelial cell function. Germline variants in *HEY2* have been linked to congenital heart defects and thoracic aneurysms, underscoring the interdependence between early developmental patterning and adult aortic wall remodeling [53]. TBX20, a T-box transcription factor essential for cardiovascular morphogenesis, also influences SMC differentiation and matrix gene expression. Variants in *TBX20* are associated with bicuspid aortic valve and ascending aortopathy, likely through disrupted signaling at the interface of valvulogenesis and aortic patterning [88].

#### 3.5.4. SMAD Nuclear Transport

Downstream of TGF-β receptor signaling, transcriptional regulation is tightly modulated by nuclear transport machinery. Importin-8 (encoded by *IPO8*) mediates the nuclear import of receptor-activated SMADs following TGF-β stimulation. Loss-of-function variants in *IPO8* impair SMAD translocation, resulting in a syndromic connective tissue disorder characterized by aortic aneurysms, skeletal anomalies, and immune dysregulation [56,57]. The nuclear localization of SMADs is a critical checkpoint in the TGF-β signaling cascade, and disruption at this level may phenocopy upstream ligand or receptor defects by attenuating ECM gene transcription.

#### 3.5.5. Post-Transcriptional Regulators

Additional transcriptional and post-transcriptional regulators play supporting roles in vascular homeostasis and oxidative defense. *HNRNPK* encodes a multifunctional RNA- and DNA-binding protein involved in chromatin remodeling, mRNA splicing, and transcriptional regulation. Although not yet directly linked to TAAD, its wide-ranging influence on gene expression may affect SMC contractile programs and cellular responses to biomechanical strain [54,55]. Prostate transmembrane protein, androgen-induced 1 (encoded by *PMEPA1*), a transmembrane protein induced by TGF-β, acts as a negative feedback regulator by promoting the degradation of activated SMADs, thus modulating downstream gene transcription. Its role in balancing profibrotic signaling suggests that it may function as a genetic modifier of aortic wall remodeling, particularly in the context of TGF-β-driven disease [77].

#### 3.5.6. Redox Regulation of Transcriptional Control

Redox regulation also contributes to nuclear and transcriptional control in the aorta. Selenocysteine insertion sequence-binding protein (encoded by *SECISBP2*) is essential for the synthesis of antioxidant selenoproteins, including glutathione peroxidases. These enzymes protect the vessel wall from oxidative damage, which can exacerbate medial degeneration and compromise ECM integrity. Impaired selenocysteine insertion sequence-binding protein function may increase vulnerability to aortic disease by tipping the balance toward reactive oxygen species accumulation and pro-inflammatory remodeling [82].

#### 3.5.7. Mediator Complex and mTOR-Linked Regulators

Finally, Mediatory complex subunit 12 (encoded by *MED12*) is a subunit of the Mediator complex, which bridges transcription factors to RNA polymerase II and regulates genes involved in cell proliferation and differentiation. While most extensively studied in tumors, mediatory complex subunit 12 also influences vascular SMC function, and knockdown studies have demonstrated defective contractility and matrix remodeling in aortic tissues [66]. Similarly, folliculin (encoded by *FLCN*) is a tumor suppressor implicated in Birt–Hogg–Dubé syndrome. Emerging evidence suggests folliculin may influence mTOR signaling and SMC behavior, with potential roles in ECM turnover and vascular remodeling [49].

#### 3.5.8. Summary and Clinical Implications

Taken together, these genes represent diverse but interconnected regulators of transcriptional output, nuclear mechanotransduction, and stress-responsive gene expression. Disruption in these systems, whether through impaired SMAD nuclear transport, altered chromatin accessibility, or oxidative imbalance, can modulate the expression of contractile proteins, ECM components, and inflammatory mediators. These changes ultimately compromise aortic wall integrity and highlight the need for comprehensive genetic screening strategies that extend beyond traditional structural or signaling genes to include the transcriptional and nuclear machinery governing vascular homeostasis.

### 3.6. Metabolic and Ion Transport

#### 3.6.1. Overview of Metabolic and Ion Transport in TAAD

Metabolic and ion transport pathways play a critical role in maintaining vascular tone, ECM remodeling, and redox homeostasis, functions essential to the integrity of the aortic wall. Genes in this category often influence TAAD through indirect mechanisms, including mineral deposition, post-translational modification, methylation, and ionic regulation of SMC contractility (refer to Table 1, Figure 2).

#### 3.6.2. Mineral Homeostasis and ECM Calcification

Regulation of mineral homeostasis and ECM calcification are key components in vascular remodeling. ATP-binding cassette, subfamily C, member 6 (encoded by *ABCC6*) is primarily expressed in the liver and kidneys and exports unknown small molecules that inhibit pathological calcification. Its deficiency causes pseudoxanthoma elasticum (PXE), characterized by abnormal mineralization of elastic fibers in the skin and arteries. In the aorta, reduced ATP-binding cassette, subfamily C, member 6 function may predispose to stiffness, elastic lamellae fragmentation, and aortic dilation [22]. Ectonucleotide pyrophosphatase/phosphodiesterase 1 (encoded by *ENPP1*) generates inorganic pyrophosphate (PPi), a physiological inhibitor of vascular calcification. Ectonucleotide pyrophosphatase/phosphodiesterase 1 deficiency results in reduced PPi availability, leading to generalized arterial calcification of infancy (GACI) and associated vascular rigidity and aneurysm formation [46]. Similarly, ATPase, Cu2+-transporting alpha polypeptide (encoded by *ATP7A*), a copper transporter essential for lysyl oxidase activity, supports crosslinking of collagen and elastin. In Menkes disease, ATPase, Cu2+-transporting alpha polypeptide dysfunction impairs ECM maturation, resulting in fragile arterial walls, tortuosity, and dissection-prone vasculature [46].

#### 3.6.3. Sulfur Amino Acid Metabolism and Epigenetic Regulation

Proper regulation of sulfur amino acid metabolism is also necessary for vascular homeostasis. *CBS* encodes cystathionine β-synthase, an enzyme involved in homocysteine metabolism. Cystathionine β-synthase deficiency results in elevated homocysteine levels (homocystinuria), a pro-inflammatory and pro-thrombotic state that damages endothelial cells and promotes vascular smooth muscle dysfunction. These pathologic processes may accelerate ECM degradation and increase risk for aortic aneurysm and dissection [33]. A closely related but distinct metabolic regulator is methionine adenosyltransferase II alpha (encoded by *MAT2A*), an enzyme that catalyzes the synthesis of S-adenosylmethionine (SAMe), the universal methyl donor used in epigenetic regulation. SAMe-dependent methylation influences DNA, RNA, protein, and lipid modifications essential for vascular gene expression and ECM remodeling. Pathogenic *MAT2A* variants can lead to aberrant methylation profiles, disrupting smooth muscle differentiation, inflammatory responses, and matrix turnover, mechanisms directly linked to aortic wall vulnerability [65].

#### 3.6.4. Ion Channel Regulation of VSMC Tone

In parallel, ionic homeostasis within SMCs is critical for contractile tone and biomechanical adaptation. *KCNN1* encodes a small-conductance calcium-activated potassium channel (SK1) that contributes to membrane repolarization in both vascular smooth muscle and endothelial cells. By regulating potassium efflux, SK1 influences intracellular calcium dynamics and thus modulates vasodilation and myogenic tone. Disrupted SK channel function may impair the vessel’s adaptive response to pulsatile stress, leading to arterial stiffness, abnormal remodeling, or predisposition to aneurysm formation [59]. These ion transport channels function as upstream regulators of mechanotransduction pathways and interact with calcium-dependent kinases and myosin light chain signaling to maintain vascular tone.

#### 3.6.5. Solute Carrier Family and Vesicular Transport

The solute carrier (SLC) family comprises a large group of membrane-bound transporters responsible for the active transport of diverse substrates, including ions, vitamins, and metabolites, across cellular membranes. While many SLCs function intracellularly, others localize to the plasma membrane and can modulate responses to ECM signals by regulating ionic gradients and redox tone. Other genes indirectly influence ECM integrity and post-translational modification through vesicular trafficking or micronutrient balance. ATPase, H+ Transporting, Lysosomal, V0 Subunit A2 (encoded by *ATP6V0A2*), a proton pump subunit localized to the Golgi apparatus, is required for proper glycosylation of ECM proteins. Variants in *ATP6V0A2* cause autosomal recessive cutis laxa type IIA, marked by vascular tortuosity and aneurysm due to aberrant ECM assembly [16]. A variant in *SLC2A10 (GLUT10)*, associated with arterial tortuosity syndrome, participates in intracellular transport of dehydroascorbic acid, affecting redox tone and TGF-β trafficking. Its loss results in medial degeneration and arterial elongation [68]. *SLC39A13 (ZIP13)* encodes a zinc transporter that regulates the function of matrix enzymes, such as lysyl hydroxylases; its disruption impairs collagen crosslinking and is associated with connective tissue fragility [84].

#### 3.6.6. Summary and Therapeutic Perspectives

Collectively, these metabolic and ion transport genes form an integrated regulatory network that modulates ECM stability, vascular tone, redox balance, and epigenetic programming. Their dysfunction can alter the structural resilience of the aortic wall both directly, via impaired matrix crosslinking and mineralization, and indirectly, through dysregulated contractility and transcriptional control. Understanding these pathways may offer novel therapeutic angles, including methylation modulators, ion channel stabilizers, or enzyme cofactor supplementation for genetically predisposed individuals.

### 3.7. Developmental and Cardiac Transcription Factors

#### 3.7.1. Overview of Developmental and Cardiac Transcription Factors in TAAD

Transcription factors that regulate cardiovascular development and SMC differentiation play a pivotal role in shaping aortic architecture and preserving vascular integrity across the lifespan. Genetic disruptions in these regulators can impair valvulogenesis, arterial remodeling, and ECM gene expression, increasing susceptibility to TAAD, particularly in the context of congenital cardiovascular anomalies (refer to Table 1, Figure 2).

#### 3.7.2. Notch Pathway Regulators

One of the most extensively studied signaling pathways involved in aortic development is the Notch pathway, which governs arterial specification, endocardial differentiation, and SMC lineage commitment. Notch Receptor 1 (encoded by *NOTCH1*), a transmembrane receptor activated by cell–cell contact, is critical for maintaining vascular identity and mediating endothelial-to-mesenchymal transition during valvulogenesis. Germline variants in *NOTCH1* have been associated with BAV and ascending aortic aneurysm, often through dysregulated elastogenesis and ECM remodeling in the proximal aorta [72,73]. Notch ligand Jagged1 (encoded by *JAG1*) complements this axis by initiating Notch signaling in adjacent cells during early vascular development. While pathogenic *JAG1* variants typically cause Alagille syndrome, recent evidence suggests that even isolated mutations can lead to nonsyndromic aortopathy through impaired SMC differentiation and structural malformations of the outflow tract [58]. The synergy between Notch Receptor 1 and Jagged1 underscores the centrality of Notch-mediated transcriptional control in shaping the embryologic origins of thoracic aortic architecture.

#### 3.7.3. TGF-β-Linked Transcriptional Repressors

Several transcription factors downstream of Notch or TGF-β signaling further refine vascular cell fate and matrix gene expression. SMAD family member 6, a known inhibitory SMAD, regulates both BMP and TGF-β pathways and serves as a transcriptional repressor that limits pro-osteogenic and profibrotic signaling in vascular cells. Loss-of-function variants in *SMAD6* are frequently associated with BAV and related aortopathy, likely through disinhibition of calcification pathways and impaired feedback regulation of TGF-β signaling [86]. SKI proto-oncogene, a potent nuclear repressor of SMAD-dependent transcription, also plays a role in the fine-tuning of TGF-β responses. Variants in *SKI* lead to Shprintzen–Goldberg syndrome, characterized by craniofacial abnormalities, connective tissue fragility, and thoracic aortic aneurysm, demonstrating how dysregulated transcriptional repression can drive both developmental and structural vascular pathology [83].

#### 3.7.4. SMC Differentiation Factors

Other transcription factors directly regulate SMC differentiation, cytoskeletal integrity, and ECM turnover. HES-related family bHLH transcription factor with YRPW Motif 2, a basic helix–loop–helix transcription factor and downstream target of Notch, controls SMC-specific gene expression and modulates endothelial plasticity. Variants in *HEY2* have been associated with both congenital heart disease and thoracic aneurysms, highlighting its dual role in cardiovascular morphogenesis and adult aortic remodeling [53]. Similarly, T-box transcription factor 20, which is essential for chamber septation and valve development, also contributes to SMC fate specification and matrix organization. Germline *TBX20* variants have been linked to BAV and aortopathy, suggesting that defects in early cardiac transcriptional programming may extend into the developing aortic wall [55].

#### 3.7.5. Chromatin-Modifying and Cytoskeletal Regulators

Expanding beyond classical transcriptional regulation, emerging factors with chromatin-modifying or cytoskeletal regulatory roles have also been implicated in vascular disease. *ABL1* encodes a non-receptor tyrosine kinase with nuclear functions, which modulates actin polymerization and intracellular signaling involved in vascular repair. Its dysregulation can alter cytoskeletal dynamics and has been associated with vascular wall instability in syndromic contexts [105]. Heterogeneous nuclear ribonucleoprotein K, a multifunctional RNA- and DNA-binding protein, regulates gene expression at multiple levels, from chromatin accessibility to mRNA processing, and may influence vascular stress responses and SMC phenotype through global transcriptional control mechanisms [54,55].

#### 3.7.6. Negative Feedback and Mediator Complex Components

In addition, prostate transmembrane protein, androgen-induced 1, an androgen-regulated transmembrane protein, functions as a negative feedback inhibitor of TGF-β signaling by promoting degradation of activated SMADs. Although primarily studied in fibrotic and oncologic settings, prostate transmembrane protein, androgen-induced 1, may influence ECM remodeling in the aorta by attenuating maladaptive TGF-β responses [77]. *MED12* encodes a component of the mediator transcriptional coactivator complex that integrates upstream transcription factor activity with RNA polymerase II function and is essential for regulating SMC contractility and proliferation. Inactivation of *MED12* leads to defective cytoskeletal remodeling and has been associated with aortic wall degeneration and dissection in functional studies [66].

#### 3.7.7. Metabolic and Mechanical Signal Integration (FLCN)

Finally, transcriptional regulators also intersect with redox-sensitive and mechanical signaling pathways that shape gene expression in vascular cells. Folliculin (encoded by *FLCN*) modulates mTOR activity and cellular metabolism. Originally implicated in Birt–Hogg–Dubé syndrome, *FLCN* has emerging roles in SMC behavior and ECM regulation, potentially linking energy sensing with aortic wall maintenance [49]. These diverse factors together form a transcriptional network that integrates developmental cues, mechanical stress, and signaling feedback to sustain arterial structure and function.

#### 3.7.8. Summary and Developmental–Vascular Intersection

Disruptions within this network, whether through altered gene expression, impaired signal transduction, or defective chromatin regulation, can result in profound structural changes to the aortic wall. The convergence of congenital cardiac malformations and vascular fragility observed in many of these genes highlights the shared developmental origins of the heart and aorta, reinforcing the need to assess transcriptional regulators not only in the context of cardiac development but also in adult-onset aortic disease.

### 3.8. Proteoglycans, Glycosylation, and Linkeropathies

#### 3.8.1. Overview of Proteoglycans in the Aortic ECM

Proteoglycans are essential structural and signaling components of the aortic ECM, where they modulate hydration, buffer mechanical load, and regulate cell–matrix interactions. Their function depends on the proper assembly of glycosaminoglycan (GAG) chains and their covalent linkage to core proteins via a tetrasaccharide linker region. Disruption of this biosynthetic sequence, whether through defective linker synthesis, impaired GAG elongation, or abnormal matrix incorporation, can weaken the arterial wall and predispose to TAAD, often in the context of connective tissue disorders, such as EDS (refer to Table 1, Figure 2).

#### 3.8.2. Linker Region Synthesis

The synthesis of the GAG–protein linkage region forms the foundation of proteoglycan structure and is catalyzed by a coordinated set of glycosyltransferases. *B3GALT6* and *B4GALT7* encode β-1,3- and β-1,4-galactosyltransferases, respectively, which initiate linker formation by sequentially attaching galactose residues to the core protein [30,31]. Defects in either gene result in spondylodysplastic EDS (spEDS), characterized by short stature, joint laxity, skin fragility, and variable vascular involvement. Beta-1,3-glucuronyltransferase (encoded by *B3GAT3*) adds the final glucuronic acid residue in the linker region, completing the assembly required for subsequent GAG chain extension [106]. Variants in *B3GAT3* cause a related spectrum of skeletal dysplasia with arterial tortuosity and aneurysm formation, emphasizing the critical role of complete linker biosynthesis in vascular matrix resilience.

#### 3.8.3. Dermatan Sulfate Chain Elongation and Sulfation

Following linker assembly, specific enzymes mediate the elongation and sulfation of dermatan sulfate chains, key GAGs in the vascular ECM. *DSE* encodes dermatan sulfate epimerase, which converts glucuronic acid to iduronic acid, a modification essential for GAG flexibility and charge density [42]. *CHST14* encodes a carbohydrate sulfotransferase, which subsequently sulfates the iduronic acid residues to stabilize the dermatan sulfate chains [35]. Variants in either gene cause musculocontractural EDS (mcEDS), marked by skin hyperextensibility, joint contractures, and arterial rupture. These structural abnormalities arise from impaired dermatan sulfate incorporation into the ECM, which reduces tissue elasticity and weakens the aortic wall’s ability to accommodate mechanical stress.

#### 3.8.4. Small Leucine-Rich Proteoglycans

Beyond biosynthesis, SLRPs such as biglycan (encoded by *BGN*), decorin (encoded by *DCN*), and lumican (encoded by *LUM*) play pivotal roles in integrating proteoglycans into the ECM and regulating TGF-β bioavailability. Biglycan, an X-linked proteoglycan, binds to collagen fibrils and modulates ECM stiffness and profibrotic signaling. Pathogenic variants in *BGN* cause a syndromic form of TAAD in males, associated with early-onset aortic dilation and systemic connective tissue findings [32]. Decorin regulates collagen fibrillogenesis by controlling fibril diameter and spacing; its loss has been shown to impair elastic fiber integrity and increase aortic dilation in animal models [41]. Lumican, another SLRP, stabilizes collagen fiber architecture and contributes to ECM hydration. Decreased *LUM* expression has been observed in acute aortic dissection specimens, suggesting a role in medial wall stability [41].

#### 3.8.5. Post-Translational Modifications in Proteoglycan Biology

Post-translational modifications further influence proteoglycan structure and ECM incorporation. *PLOD1* and *PLOD3* encode lysyl hydroxylases that hydroxylate and glycosylate lysine residues in collagen and proteoglycans. These modifications facilitate stable crosslinking and glycan addition. While *PLOD1* variants cause kyphoscoliosis EDS, *PLOD3* variants result in a Stickler-like syndrome with vascular fragility, skeletal abnormalities, and defective ECM glycosylation [14,76]. The role of Procollagen-Lysine, 2-oxoglutrate 5-dioxygenase 3 in both hydroxylation and glycosylation underscores its importance in proteoglycan-collagen interface biology.

#### 3.8.6. Transcriptional Regulators of ECM and Proteoglycan Genes

Transcriptional control of proteoglycan synthesis and ECM assembly is modulated by proteins produced by genes such as *PRDM5* and *ZNF469*, both of which are associated with brittle cornea syndrome. *PRDM5* encodes a zinc finger transcription factor that regulates expression of ECM genes, including collagens and proteoglycans [78], while zinc finger protein 469 influences ECM composition and corneal transparency, but it has also been linked to aortic and arterial aneurysmal disease [78]. Disruption of either gene can lead to abnormal matrix composition and impaired structural integrity of the aortic wall, expanding the relevance of brittle cornea-associated genes to systemic arteriopathies.

#### 3.8.7. Summary and Clinical Implications

Together, these genes define a linear and cooperative pathway that begins with proteoglycan core attachment, continues through GAG chain biosynthesis and sulfation, and culminates in ECM incorporation and transcriptional maintenance. Variants at any step can compromise the mechanical and biochemical properties of the vascular wall, rendering it vulnerable to dissection or rupture. As a group, these genes underscore the importance of proteoglycan biology not only in rare syndromic forms of EDS but also in broader vascular pathology and suggest that linkeropathies may be under-recognized contributors to hereditary TAAD.

### 3.9. Emerging Genes and Hypothesis-Generating Candidates

#### 3.9.1. Overview of Emerging Gene Candidates in TAAD

Although substantial progress has been made in identifying monogenic causes of TAAD, an increasing number of mechanistically plausible genes have emerged from transcriptomic, proteomic, and candidate-based studies. While not yet formally established as TAAD-causative by genetic linkage or large cohort validation, these genes converge on biological processes central to vascular stability, including ECM regulation, SMC adhesion, cytoskeletal dynamics, and transcriptional programming. Their integration into known aortopathy pathways provides a rationale for further investigation and possible inclusion in expanded gene panels (refer to Table 1, Figure 1, Figure 2, Figure 3 and Figure 4).

#### 3.9.2. Proteoglycan Remodeling and ECM Regulation

Several emerging candidates influence ECM organization and proteoglycan matrix remodeling. Decorin and lumican are SLRPs that bind to collagen fibrils, regulate their diameter, and interact with TGF-β signaling components. Knockout models of *DCN* exhibit elastic fiber disorganization and increased susceptibility to aortic dilation, while reduced *LUM* expression has been observed in aneurysmal human tissue [41,64]. These findings suggest that altered SLRP function may weaken matrix integrity and amplify profibrotic signaling, creating a permissive environment for aortic degeneration.

#### 3.9.3. TGF-β Pathway Modulators

Regulators of TGF-β signaling, such as prostate transmembrane protein, androgen-induced 1, also warrant attention. *PMEPA1* encodes a TGF-β-inducible transmembrane protein that limits pathway activation by promoting degradation of receptor-activated SMADs. Downregulation of *PMEPA1* may contribute to sustained TGF-β signaling and maladaptive ECM remodeling, particularly in genetically or hemodynamically primed aortic tissue [107]. Though best studied in oncology, its role as a modulator of fibrotic tone and matrix turnover positions it as a candidate disease modifier in TGF-β-driven arteriopathies.

#### 3.9.4. Cytoskeletal and Cell–Cell Adhesion Proteins

Disruption of cytoskeletal tension and cell–cell adhesion also features prominently among emerging TAAD-associated genes. *CTNNA1* and *CTNNB1*, encoding α- and β-catenin, respectively, mediate adherens junction stability and link cadherins to the actin cytoskeleton. These proteins maintain intercellular cohesion under mechanical strain and participate in Wnt signaling, a pathway increasingly implicated in aneurysmal remodeling [39,108]. Their binding partner, N-cadherin 2, is essential for SMC-SMC adhesion and survival; dysregulation of this axis disrupts cytoskeletal anchoring and promotes phenotypic switching, a hallmark of aneurysm-prone medial degeneration [34].

#### 3.9.5. Focal Adhesion Integrity

Further integrating this structural network are focal adhesion proteins, such as talin-1 and vinculin, which couple integrins to the actin cytoskeleton and transmit mechanical force across the cell–ECM interface. Downregulation of *TLN1* in dissected aortic tissue correlates with increased SMC migration and proliferation, while loss of *VCL* expression has been documented in acute aortic dissection, suggesting a shared role in maintaining aortic wall mechanostability [26,96]. These findings support the hypothesis that focal adhesion integrity is critical to preventing vascular rupture under physiological load.

#### 3.9.6. Desmosomal Proteins with Vascular Roles

Desmosomal proteins traditionally linked to cardiac arrhythmia are now recognized for their potential roles in vascular tissue cohesion. Plakophilin-2 and desmoplakin are core structural components of desmosomes that tether intermediate filaments between adjacent cells. While variants in both genes are well-established in arrhythmogenic cardiomyopathy, recent evidence suggests that they may also affect vascular SMC anchoring and resistance to cyclic stress [43,75]. Given their dual expression in cardiac and vascular tissues, these genes exemplify how overlapping cardiac–vascular phenotypes may arise from shared cytoskeletal vulnerabilities.

#### 3.9.7. Gap Junction Communication in Aortic Wall Cohesion

Endothelial communication and coordination of vascular tone rely on gap junction proteins, such as connexin 37 (encoded by *GJA4*) and connexin 40 (encoded by *GJA5*). These channels facilitate ionic and metabolic exchange between endothelial cells and between endothelium and underlying SMCs. Disruption of connexin-mediated signaling can impair vessel wall homeostasis, promote inflammation, and facilitate ECM remodeling, all of which contribute to aneurysm susceptibility [40,52]. The role of *GJA5* in hypertension further highlights the relevance of endothelial–SMC cross-talk in regulating mechanical stress responses in the aortic wall.

#### 3.9.8. Transcriptional Regulators with Vascular Implications

Transcriptional regulators, such as zinc finger, BTB-containing protein 20 (encoded by *ZBTB20*), and zinc finger homeobox 3 (encoded by *ZFHX3*), represent additional candidates that may modulate aortic structure through epigenetic or developmental programming. Zinc finger and BTB-containing protein 20 influence PI3K and MAPK signaling pathways and have been linked to vascular remodeling and SMC phenotypic modulation [97]. *ZFHX3*, a zinc finger homeobox gene associated with atrial fibrillation and neurodevelopmental disorders, is expressed in vascular tissues and may regulate cytoskeletal gene expression and inflammatory pathways relevant to medial degeneration [98]. Though primarily studied in neurologic and cardiac contexts, their emerging vascular expression profiles and regulatory functions make them plausible candidates for syndromic or mosaic arteriopathy.

#### 3.9.9. Summary and Clinical Implications

Taken together, these emerging genes reflect a spectrum of biologically plausible contributors to TAAD pathogenesis. Though not yet confirmed as monogenic causes, their integration into known mechanistic frameworks, ECM remodeling, SMC contractility, adhesion signaling, and transcriptional regulation underscores their potential relevance. Further validation through high-throughput sequencing, single-cell profiling, and mechanistic studies will be essential to clarify their roles and inform the development of more comprehensive diagnostic panels for hereditary aortic disease.

### 3.10. Overlap Syndromes and Dual-Phenotype Genes

Certain genes implicated in TAAD display phenotypic overlap with congenital heart disease, neurodevelopmental disorders, and syndromes classically categorized outside of connective tissue frameworks. These dual-phenotype genes underscore the developmental and functional interdependence between the vasculature and other organ systems, often reflecting shared pathways in embryologic signaling, cytoskeletal dynamics, or transcriptional regulation (refer to Table 1, Figure 1, Figure 2, Figure 3 and Figure 4).

#### 3.10.1. Developmental Regulators Linking Cardiac Morphogenesis and Aortic Integrity

A subset of these genes highlights the connection between aortic disease and congenital heart malformations through disruption of key developmental regulators. *NOTCH1* encodes a transmembrane receptor integral to endothelial–mesenchymal signaling during cardiac valve formation. Pathogenic variants in *NOTCH1* have been associated with BAV and calcific aortic valve disease and may also promote aneurysmal degeneration through impaired vascular differentiation or excess calcification [73]. Similarly, *JAG1*, encoding a Notch ligand, is best known for its role in Alagille syndrome, but recent data suggest that individuals with *JAG1* variants may also harbor isolated aneurysmal disease without overt syndromic features [58]. Both genes participate in the Notch signaling axis, a pathway critical for cell fate determination in vascular SMCs and cardiac precursors. Disruption of this pathway can alter the balance between contractile and synthetic phenotypes in SMCs, a process central to medial degeneration.

#### 3.10.2. Transcription Factors Bridging CHD and Arteriopathy

Additional transcription factors, such as HES-related family bHLH transcription factor with YRPW Motif 2 (HEY2) and T-box transcription factor 20, further bridge the gap between congenital heart defects and arteriopathy. HEY2 is a transcriptional effector downstream of Notch signaling that governs SMC proliferation and regional identity during aortic development. Germline variants in *HEY2* have been associated with both congenital heart disease and thoracic aortic aneurysms, suggesting a shared developmental etiology [53]. T-box transcription factor 20 regulates chamber-specific myocardial differentiation and has now been implicated in familial BAV with aortic dilation [55]. Together, these findings point to the recurring theme that genes regulating cardiac morphogenesis may also influence proximal aortic integrity.

#### 3.10.3. Nuclear Transport and Chromatin Regulators

Beyond developmental transcription factors, components of nuclear transport and chromatin regulation have also emerged as dual-phenotype contributors. Importin-8 mediates nuclear import of regulatory proteins and is essential for proper nuclear signaling in both immune and cardiovascular systems. Biallelic loss-of-function variants in *IPO8* result in syndromic thoracic aortic aneurysm with immune dysregulation and skeletal anomalies, supporting the notion that impaired nucleocytoplasmic transport can have broad developmental consequences, including on aortic structure [53]. Similarly, zinc finger homeobox 3, a zinc finger transcription factor involved in neuronal differentiation and atrial fibrillation susceptibility, has been identified in patients with intellectual disability and vascular anomalies [98]. These genes highlight how perturbations in nuclear regulatory machinery may contribute simultaneously to neurodevelopmental and vascular phenotypes through altered gene expression programs.

#### 3.10.4. Cytoskeletal and Signaling Proteins

Cytoskeletal and signaling proteins also play a central role in syndromes that span the neurologic and vascular systems. Filamin A crosslinks actin filaments and participates in mechanosensitive signaling within the vascular wall. In addition to causing periventricular nodular heterotopia in the brain, *FLNA* variants are associated with aortic dilation, emphasizing its dual role in neural migration and vascular integrity [49]. Similarly, *MED12*, which encodes a component of the Mediator complex involved in transcriptional activation, regulates SMC function and has been implicated in both aortic dissection and developmental syndromes with intellectual disability and craniofacial dysmorphism [66]. These proteins operate at the interface of cytoskeletal architecture and gene regulation, suggesting that mechanical signaling defects can manifest across multiple organ systems.

#### 3.10.5. Mitochondrial and Redox Pathway Genes

Lastly, the mitochondrial and oxidative stress response gene *SELENOT* (encodes selenoprotein T) has emerged in whole-exome studies of syndromic TAAD. Deficiency of selenoproteins, including *SELENOT*, impairs redox homeostasis and protein folding, particularly in the cardiovascular system. Recent work links *SELENOT* variants to aortic aneurysm formation in mouse models, with associated metabolic and neurologic phenotypes in humans [82]. This observation expands the concept of syndromic aortopathy to include redox dysregulation and cellular stress adaptation as possible unifying mechanisms across disparate tissue types.

#### 3.10.6. Summary and Clinical Implications

In sum, these overlapping genes highlight the importance of developmental, transcriptional, and cytoskeletal integrity in both vascular and extracardiovascular tissues. Their phenotypic breadth reflects shared embryologic origins and reinforces the value of broad systemic evaluation in patients presenting with TAAD, especially when syndromic features or extracardiac manifestations are present. Continued investigation into these dual-phenotype genes will refine genotype–phenotype correlations and may improve early recognition of at-risk individuals through interdisciplinary screening.

## 4. Clinical Translation and Future Directions

### 4.1. Diagnostic Yield and Gene-Specific Patterns

The expanding spectrum of genetic contributors to TAAD has substantial implications for clinical management and risk stratification. Historically, genetic testing focused on core syndromic genes, such as *FBN1*, *TGFBR1/2*, and *COL3A1*, which collectively account for the majority of diagnoses in classic connective tissue disorders like Marfan syndrome, LDS, and vEDS [11,12,13,14]. The contribution of specific genes to heritable TAAD varies by inheritance pattern and phenotype. Canonical syndromic genes, such as *FBN1* and *COL3A1,* are highly penetrant and account for the majority of Marfan syndrome and vascular Ehlers–Danlos cases, respectively. Common nonsyndromic genes, including *ACTA2* and *MYH11*, collectively explain a substantial proportion of familial TAAD but occur less frequently than syndromic causes. Rare high-penetrance variants in genes such as *MYLK*, *LOX*, and *PRKG1* are typically identified in small pedigrees or isolated cases, often with aggressive disease at smaller aortic diameters. Many emerging or putative modifier genes have uncertain prevalence and penetrance, underscoring the need for cautious interpretation until larger cohort data become available. However, as this review illustrates, a broader array of genes, including those involved in ECM organization, TGF-β modulation, SMC cytoskeletal integrity, and cell–cell junctions, contribute meaningfully to disease expression across syndromic, nonsyndromic, and overlapping phenotypes (refer to Table 1, Figure 1, Figure 2, Figure 3 and Figure 4) [24,109,110,111].

Multigene panel testing now yields diagnostic rates approaching 30% in individuals with suspected hereditary TAAD [2]. However, this yield may be underestimated in patients presenting with attenuated or atypical syndromic features. While most individuals with Marfan syndrome, LDS, or vEDS exhibit both cardiac and extracardiac manifestations, milder phenotypes may obscure clinical suspicion. Broadening gene panels to include underrecognized contributors, such as *PLOD3* [76], *SPARC* [78], *IPO8* [56,57], *ZBTB20* [97], and *TLN1* [95], may help capture additional diagnoses, although the extent of added yield remains to be fully established. Gene-informed approaches can also guide surgical timing; for example, *ACTA2* variants are associated with increased dissection risk at smaller aortic diameters [25,69], whereas some *TGFB1*, *TGFB2*, or *FBN1* variants may follow more indolent courses [111].

### 4.2. Gene-Informed Surgical and Surveillance Strategies

An equally important frontier is the stratification of surveillance intensity and intervention thresholds based on molecular diagnosis. While current guidelines recommend early and frequent imaging in patients with known pathogenic variants, they are not universally gene-agnostic; specific recommendations exist for genes, such as *FBN1*, *TGFBR1*, and *ACTA2*, among others [19]. However, many cases remain without gene-specific guidance [111]. Emerging genotype–phenotype correlations suggest that personalized surveillance intervals, potentially modulated by additional risk factors, such as hypertension, family history, or sex, may soon become feasible. This is particularly pertinent in genes with variable penetrance or dual-phenotypic overlap, such as *FLNA* (neurovascular/cardiac), *DSP* (aortic/cardiomyopathy), or *PRDM5* (ocular/aortic) [43,50,78].

### 4.3. From Variant Function to Therapeutic Targeting

Beyond diagnostics, understanding the functional consequences of pathogenic variants holds promise for targeted therapeutics. For instance, prostate transmembrane protein, androgen-induced 1, an endogenous inhibitor of SMAD signaling, may act as a modifiable brake on TGF-β-mediated ECM dysregulation, positioning it as a potential therapeutic axis in TGF-β-driven aortopathies [107]. Similarly, gene-encoding SLRPs, like *DCN* and *LUM*, both found to be downregulated in aneurysmal tissues, may eventually serve as biomarkers or therapeutic targets for matrix stabilization [41,64]. The identification of such modulators also creates an opportunity for polygenic risk scoring, especially in patients with nonsyndromic or borderline presentations.

Therapeutic approaches for heritable TAAD are evolving toward pathway-specific interventions informed by genetic diagnosis. In TGF-β-driven syndromes, such as Marfan and Loeys–Dietz, angiotensin receptor blockers (e.g., losartan) have shown efficacy in attenuating aortic root growth, with ongoing trials evaluating optimal dosing and long-term outcomes [109]. Preclinical studies in Loeys–Dietz mouse models have demonstrated benefit from mTOR inhibition, suggesting a potential role for agents, such as sirolimus, in severe or refractory disease [112]. For metabolic and epigenetic regulators, experimental inhibition of MAT2A-dependent methylation has been shown to normalize aberrant transcriptional profiles in vascular smooth muscle cells, while targeted modulation of KCNN1 is being explored to restore vascular tone and mechanosensing [59,65]. Redox-based strategies, including selenium supplementation in SECISBP2 or SELENOT deficiency, may help mitigate oxidative injury to the aortic wall [82]. While many of these approaches remain in early stages, they illustrate the potential for genotype-guided therapy to complement current surgical and surveillance-based management paradigms.

### 4.4. Expanded Phenotyping for Comprehensive Risk Assessment

The frontier of clinical translation will also require refined phenotyping. Traditional imaging confined to the aortic root may miss critical disease in the distal aorta, cerebral vasculature, or systemic arteries. In some cases, the extent of imaging can be gene-based; for example, head-to-pelvis vascular imaging is recommended for individuals with *TGFBR1* or *TGFBR2* variants, given their propensity for diffuse arteriopathy [111]. Whole-body vascular imaging and functional assays, such as pulse wave velocity, dynamic flow MRI, or skin biopsy for ECM assessment, may further unmask subclinical disease, especially in genes with variable expression such as *B4GALT7*, *ACTN2*, or *IPO8* [31,57,103].

### 4.5. Translational Parallels in Peripheral Artery Disease (PAD)

The mechanistic overlap between TAAD and other large- and medium-vessel arteriopathies suggests that genetic insights gained in TAAD could also inform the management of conditions, such as peripheral artery disease (PAD). Shared processes—such as extracellular matrix degradation, smooth muscle dysfunction, and maladaptive vascular remodeling—may influence susceptibility, disease progression, and response to therapy in both disorders. Integrating genetic risk stratification into PAD management could enable earlier detection of high-risk individuals, personalized surveillance intervals, and targeted interventions, paralleling approaches increasingly applied in TAAD. This concept aligns with the framework proposed by Chioncel et al., which emphasizes precision-based medical management in PAD [113].

### 4.6. Genetic Differential Diagnosis and Overlap with Hypertrophic Cardiomyopathy

Emerging evidence suggests that thoracic aortic dilation may occur in the context of other heritable cardiovascular diseases, notably hypertrophic cardiomyopathy (HCM), with a recent cohort study reporting a 9.0% prevalence of dilated aorta among HCM patients [114]. While the underlying mechanisms remain unclear, potential explanations include shared genetic pathways influencing sarcomeric proteins, extracellular matrix remodeling, and vascular smooth muscle integrity. These findings highlight the importance of genetic differential diagnosis when encountering aortic dilation in patients with overlapping or atypical phenotypes. Lifestyle factors may also contribute, as large imaging studies have reported variations in aortic size in relation to regular exercise in otherwise healthy populations [115,116]. Integrating genetic results with clinical context, family history, and lifestyle assessment will be critical to avoid misclassification and to guide tailored surveillance strategies.

### 4.7. Family-Based Screening and Gene Panel Curation

Finally, the importance of early detection in at-risk relatives cannot be overstated. Family-based cascade screening remains one of the most cost-effective strategies in genetic aortic disease [110], yet its success hinges on the comprehensiveness of initial gene panels and the awareness of the broader genetic landscape. While integrating emerging genes into clinical algorithms and educating both clinicians and patients about dual-phenotype overlap may help close diagnostic and preventive gaps, this approach is not without limitations. Many newly reported genes lack definitive evidence for disease causality and are classified as genes of uncertain significance. Including variants in such genes may generate background noise in genetic reports, potentially misleading patients, families, and providers. Therefore, gene panels should be thoughtfully curated, and genetic reports must clearly outline the strength of evidence and limitations associated with any findings.

## 5. Conclusions

The genetic architecture of thoracic aortic aneurysms and dissections (TAAD) is far more diverse than previously recognized, spanning classical syndromic genes, genes governing extracellular matrix integrity and TGF-β signaling, cytoskeletal and junctional stability, proteoglycan biology, transcriptional regulation, and emerging functional candidates. While foundational genes like *FBN1*, *TGFBR1*, and *ACTA2* remain critical to early diagnosis and management, our expanding understanding of novel and overlapping contributors, including *IPO8*, *SPARC*, *ZBTB20*, and multiple linkeropathy genes, points to the possibility of enhanced genetic testing and phenotypic surveillance.

It is the hope that the integration of novel molecular insights into routine clinical care can enhance diagnostic accuracy, improve familial risk assessment, and ultimately inform gene-specific management strategies. Continued discovery and mechanistic validation of emerging candidates will be essential, not only to refine genotype–phenotype correlations but also to guide the development of targeted therapies in genetically predisposed individuals. A multidisciplinary, genomically informed strategy will be central to reducing morbidity and mortality across the spectrum of heritable aortic disease.

## Figures and Tables

**Figure 1 medsci-13-00155-f001:**
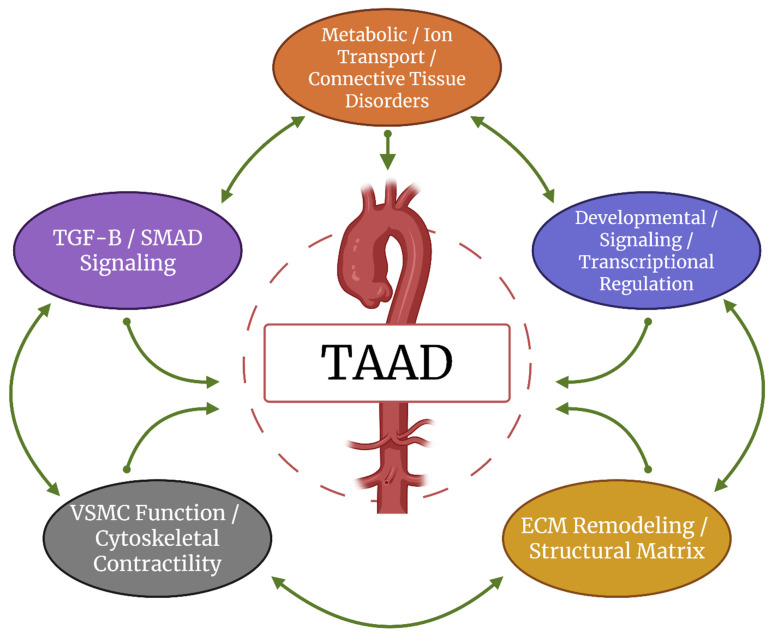
TAAD pathway integration network. Schematic representation of the major mechanistic groups contributing to TAAD. Each oval represents a functional category of genes implicated in TAAD: TGF-β/SMAD signaling, extracellular matrix (ECM) remodeling/structural matrix, vascular smooth muscle cell (VSMC) function/cytoskeletal contractility, metabolic/ion transport/connective tissue disorders, and developmental/signaling/transcriptional regulation. Arrows indicate the direction of influence on TAAD pathogenesis, with green solid arrows representing processes that accelerate or promote disease. The diagram illustrates that each pathway can independently contribute to TAAD while also engaging in cross-talk with other mechanistic groups, establishing a network of interconnected processes that collectively drive aortic wall weakening and progression to aneurysm or dissection [3].

**Figure 4 medsci-13-00155-f004:**
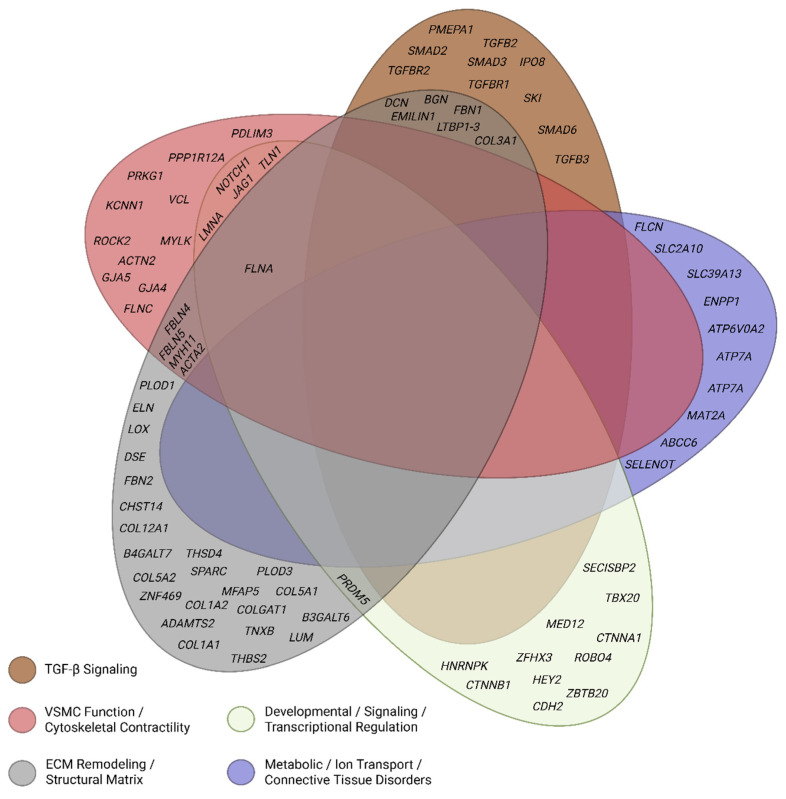
Functional overlap of genes implicated in heritable TAAD. This Venn diagram illustrates the distribution and overlap of genes across five major functional categories relevant to TAAD pathogenesis: TGF-β signaling, vascular smooth muscle cell (VSMC) function and cytoskeletal contractility, extracellular matrix (ECM) remodeling and structural integrity, metabolic/ion transport and connective tissue disorders, and developmental or transcriptional regulation. Many genes intersect multiple pathways, reflecting the complex, pleiotropic mechanisms underlying aortic wall vulnerability. Shared nodes, such as FLNA, FBN1, TGFB2, and ZBTB20, highlight genes with multifunctional roles that integrate signaling, structural, and regulatory functions in the pathophysiology of TAAD [104].

## Data Availability

No new data created.

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
