# Peer review of "The Expanding Genetic Architecture of Arteriopathies: From Canonical TAAD Genes to Emerging Connective Tissue and Signaling Pathways"

_medsci, 2025, doi:10.3390/medsci13030155_

Round 1

Reviewer 1 Report

Comments and Suggestions for Authors

Thoracic aortic aneurysm and dissection may include today more than 75 genes implicated in pathogenesis, categorizing them according to major mechanistic roles . This paper is extremely interesting focusing on the hope that  the integration of novel molecular insights into routine clinical care may  enhance diagnostic accuracy, improve familial risk assessment, and ultimately inform gene-specific management strategies. I have only minor comment in order to improve the manuscript . Today special emphasis is placed on emerging genes with variable or overlapping clinical phenotypes: in this scenario authors should also describe potential genetic differential diagnosis. In particular, some authors reported a novel observation with 9.0% prevalance of dilated aorta in HCM patients. Further studies are needed to help define the genetic and pathophysiologic basis as well as the clinical implications of this association. Therefore, authors should better discuss about this important point,  also considering the potential impact of regular exercise practice in asymptomatic healthy population   (DOI: 10.1093/ehjci/jew292; and DOI: 10.1016/j.echo.2020.11.003; DOI: 10.1016/j.ijcard.2021.10.013) Please cite 3 suggested references and amplify discussion

Reviewer 2 Report

Comments and Suggestions for Authors

The article provides a thorough and well-organised review of the genetic basis of thoracic aortic aneurysms and dissections (TAAD), effectively synthesising the roles of both established genes (e.g., FBN1, TGFB2, TGFB3, ACTA2) and emerging contributors, such as those involved in metabolic pathways, ion transport, and transcriptional regulation (e.g., SLC2A10, ZBTB20, HRNPK). The detailed exploration of molecular pathways, particularly TGF-β signalling and its impact on extracellular matrix (ECM) stability, vascular tone, and smooth muscle cell (SMC) phenotype, is a highlight. The inclusion of a comprehensive gene classification table (Table 1) and the central figure (Figure 1) effectively consolidates the functional and clinical significance of these genetic factors, making the review a valuable resource for researchers and clinicians. The discussion on clinical applications, such as gene-informed surgical timing and surveillance strategies, is insightful and underscores the translational potential of these findings. The acknowledgement of limitations in current genetic testing panels and the need for further validation of novel genes like IPO8 and SPARC enhances the article’s scholarly rigour.

**Suggestions for Improvement:**

  1. **Clarity in Pathway Integration**: The article could improve by better integrating the discussed pathways (e.g., TGF-β, ECM synthesis, ion transport). A schematic or expanded figure showing how these pathways interact to contribute to aortic pathology would enhance clarity, especially for readers less familiar with molecular mechanisms.

  1. **Quantification of Genetic Contributions**: Including a table or brief discussion on the relative contribution of each gene or pathway to TAAD risk (e.g., penetrance, variant prevalence) would help prioritise clinically actionable genes.

  1. **Therapeutic Horizons**: The brief mention of potential therapies (e.g., methylation modulators, ion channel stabilisers) is promising but underdeveloped. Expanding this section with specific examples from ongoing research or clinical trials would strengthen the article’s relevance to precision medicine.

  1. **Peripheral Artery Disease (PAD) Management**: Given the overlap between TAAD and other arteriopathies, the article could benefit from a brief discussion on how genetic insights from TAAD might inform the management of related conditions, such as peripheral artery disease (PAD). For instance, integrating genetic risk stratification into PAD management could guide personalised surveillance and intervention strategies, similar to those proposed for TAAD. A relevant reference to support this discussion is the article by Chioncel et al (DOI: 10.1097/MJT.0000000000000916), which reviews medical management strategies for PAD and could be cited to highlight parallels in genetic and clinical approaches.

  1. **Formatting and Accessibility**: Some sections, particularly those on less-studied genes (e.g., ZBTB20, DSE), could be streamlined to reduce redundancy and improve readability. Subheadings for each gene or pathway would help navigate the dense content. Additionally, the manuscript contains OCR-related artifacts (e.g., repeated numbers, fragmented text) that should be meticulously corrected to ensure publication quality.

Overall, this review is a robust contribution to the field of vascular genetics, effectively bridging molecular research with clinical practice. Congrats to the authors!

Round 2

Reviewer 1 Report

Comments and Suggestions for Authors

Manuscript improved. Congratulations